# Shared functional organization between pulvinar-cortical and cortico-cortical connectivity and its structural and molecular imaging correlates

**Gianpaolo Antonio Basile**[1]**, Augusto Ielo**[2]**, Lilla Bonanno**[2]**, Antonio Cerasa**[3,4,5]**, Giuseppe Santoro**[1]**, Demetrio Milardi**[1]**, Giuseppe Pio Anastasi**[1]**, Ambra Torre**[1]**, Sergio Baldari**[6]**, Riccardo Laudicella**[6]**, Michele Gaeta**[7]**, Marina Quartu**[8]**, Maria Pina Serra**[8]**, Marcello Trucas**[8]**, Angelo Quartarone**[2]**, Manojkumar Saranathan**[9]**, Alberto Cacciola**[10,11]*****

[1]Brain Mapping Lab, Department of Biomedical, Dental Sciences and Morphological and Functional Imaging, University of Messina, Messina, Italy; [2]IRCCS Centro Neurolesi 'Bonino Pulejo', Messina, Italy; [3]Institute for Biomedical Research and Innovation (IRIB), National Research Council of Italy (CNR), Messina, Italy; [4]S. Anna Institute, Crotone, Italy; [5]Pharmacotechnology Documentation and Transfer Unit, Preclinical and Translational Pharmacology, Department of Pharmacy, Health Science and Nutrition, University of Calabria, Arcavacata, Italy; [6]Nuclear Medicine Unit, Department of Biomedical, Dental Sciences and Morphological and Functional Imaging, University of Messina, Messina, Italy; [7]Radiology Unit, Department of Biomedical, Dental Sciences and Morphological and Functional Imaging, University of Messina, Messina, Italy; [8]Section of Cytomorphology, Department of Biomedical Sciences, University of Cagliari, Cittadella Universitaria di Monserrato, Monserrato, Italy; [9]Department of Radiology, University of Massachusetts Chan Medical School, Worcester, United States; [10]Department of Biomedical Sciences, Humanitas University, Milan, Italy; [11]IRCCS Humanitas Research Hospital, Milan, Italy

*For correspondence: alberto.cacciola0@gmail.com

Competing interest: The authors declare that no competing interests exist.

## eLife Assessment

This study presents a **useful** characterisation of the topographical organisation of the human pulvinar, an associative thalamic subregion crucial for visual perception and attention. The evidence supporting the conclusions is **solid** given the multimodal validation and replication across datasets, although even higher-resolution imaging data would have strengthened the study. In their revised manuscript, the authors elaborated further on the motivation for their study and conducted several robustness checks. Nevertheless, there remains an opportunity for a more fully integrated interpretation of the findings. The work would be of interest to neuroscientists, neurologists, and neuropsychiatrists working on pulvinar functioning in health and disease.

**Abstract** The pulvinar, the largest thalamic nucleus, is a highly interconnected structure supporting perception, visuospatial attention, and emotional processing. Such a central role relies on a precise topographical organization reflected in anatomical connectivity and neurochemical markers. Traditionally subdivided into distinct subnuclei, recent work shows that these divisions only partially explain its organization, which is better captured by continuous gradients of cortical

connections along dorso-ventral and medio-lateral axes. While well studied in primates, this gradient-based architecture remains less explored in humans. The present work combines high-quality, multimodal structural and functional imaging with a whole-brain, large-scale, PET atlas mapping 19 neurotransmitter systems. By applying diffusion embedding to tractography, functional connectivity, and receptor coexpression, we identify multiple gradients of structural connections, functional coactivation, and molecular binding patterns. These converge on a shared representation along the dorso-ventral and medio-lateral axes of the human pulvinar, aligning with connectivity transitions from lower-level to higher-order cortical regions. Moreover, this is paralleled by gradual changes in the expression of molecular markers associated with key neuromodulator systems, including serotoninergic, noradrenergic, dopaminergic, and opioid systems. Our findings advance the understanding of pulvinar anatomy and function, offering an exploratory framework to investigate the role of this structure in both health and disease.

## Introduction

The pulvinar complex stands out as the largest nucleus within the human thalamus, serving as a pivotal hub in a myriad of cortico-subcortical networks that intricately interconnect various cortical regions of the brain (*Benarroch, 2015*). Anatomical studies in primate brains have highlighted its primary sources of afferent and efferent connections, primarily stemming from the primary and secondary visual areas. However, its connectivity also extends to other crucial regions such as the temporal lobe, primary sensory, prefrontal, and cingulate cortices (*Pons and Kaas, 1985*; *Romanski et al., 1997*; *Lyon and Kaas, 2002*; *Fang et al., 2006*). Through its extensive interplay with cerebral cortical areas, the pulvinar is believed to play a pivotal role in orchestrating integrative processes crucial for context-dependent modulation of visuospatial attention (*Jaramillo et al., 2019*; *Fiebelkorn and Kastner, 2020*). It has been postulated that the pulvinar mediates the selection of relevant information from the environment by generating alpha oscillations, thereby potentially modulating neuronal gain in thalamocortical circuits (*Bourgeois et al., 2020*). Nonetheless, despite the acknowledged significance of cortico-pulvinar connectivity in these theoretical frameworks, the functional and anatomical organization of this network in the human brain remains relatively underexplored.

Traditionally, the primate pulvinar complex has been anatomically divided into four subnuclei: the anterior (oral), the inferior, the lateral, and the medial pulvinar nuclei (*Olszewski and Baxter, 1981*). While this subdivision has long served as a foundational framework for understanding pulvinar organization, anatomical investigations in nonhuman primates have revealed that connectivity patterns extend beyond discrete cytoarchitectonic subdivisions (*Lysakowski et al., 1988*; *Webster et al., 1993*; *Adams et al., 2000*). Instead, accumulating evidence suggests a spatially continuous organization in space, wherein multiple representations of the cortical sheet and visual fields coexist (*Shipp, 2001*). These representations, often termed 'maps', adhere to the 'replication principle', wherein densely connected cortical regions exhibit overlapping representations within the pulvinar (*Shipp, 2003*). Moreover, these representations are loosely associated with chemoarchitectural domains (*Gutierrez and Cusick, 1997*; *Stepniewska and Kaas, 1997*; *Gutierrez et al., 2000*).

Translating our understanding of the functional and anatomical organization of the pulvinar complex to the human brain has proven challenging due to the small size of the underlying structures and the inherent limitations of imaging techniques. Consequently, there has been a relative scarcity of experimental works investigating pulvinar structure and function in vivo in the human brain. While functional and structural neuroimaging techniques have been successfully employed to explore the anatomy and connectivity of the human pulvinar (*Leh et al., 2008*; *Tamietto et al., 2012*; *Arcaro et al., 2015*; *Arcaro et al., 2018*; *Mai and Majtanik, 2019*; *Basile et al., 2021*), only a few studies have delved into the connectional topography of this structure. Functional MRI (fMRI)-based parcellation studies, employing task-based activation or resting-state connectivity fingerprinting, have provided insights into functional specialization within subregions of the human pulvinar (*Barron et al., 2015*; *Arcaro et al., 2018*; *Guedj and Vuilleumier, 2020*). These investigations confirm that, akin to findings in nonhuman primates, functional regions within the human pulvinar are loosely associated with the anatomical subdivision into nuclei. However, relying on voxel-wise clustering methods, these studies share the common assumption of discrete connectivity units and therefore may not fully capture the spatially continuous and transient nature of pulvinar connectivity patterns.

Recently, gradient mapping techniques have emerged as a valuable tool for investigating the structure-function relationship in the human brain. These techniques utilize dimensional decomposition algorithms, such as diffusion embedding (*Coifman and Lafon, 2006*), to map high-dimensional to manifold low-dimensional brain features, known as 'gradients'. These gradients are commonly interpreted as representing spatial patterns of smooth transitions in biological features of interest, either within or across brain structures (*Haak et al., 2018*; *Hong et al., 2020*). Over the past decade, gradient mapping methods have been extensively applied to various biological features, including structural and functional connectivity, structural covariance, MRI-based microstructural measures, receptor, and gene expression (*Bajada et al., 2017*; *Paquola et al., 2020*; *Paquola et al., 2019a*; *Paquola et al., 2019b*; *Valk et al., 2020*; *Vos de Wael et al., 2021*; *Hansen et al., 2022b*). These methods have been employed to explore the functional anatomy of the cerebral cortex (*Margulies et al., 2016*), as well as specific cortical or subcortical areas, such as the striatum, thalamus, hippocampus, and cerebellum (*Guell et al., 2018*; *Przeździk et al., 2019*; *Tian et al., 2020*; *Yang et al., 2020*; *Müller et al., 2020*; *Katsumi et al., 2023*). In particular, the existing investigations of thalamic connectivity within the gradient framework have revealed general organizational principles within the thalamus that are partially reflected in thalamic cytoarchitecture subdivision and have been related to core and matrix thalamic neuronal subpopulation and to their differential contribution to large-scale connectivity networks (*Yang et al., 2020*; *Müller et al., 2020*). However, given the remarkable functional multiplicity of the pulvinar complex, it is possible that these global spatial organization patterns only partly account for the full complexity and richness of pulvinar functional topography. With this in mind, isolating pulvinar connectivity from the remaining thalamocortical connectome would ensure that local organizational principles are not obscured by the global connectotopy of the entire thalamus.

The present study aims to investigate the gradient organization of the human pulvinar using a multimodal approach. By leveraging high-quality structural, diffusion-weighted, and resting-state functional MRI (rs-fMRI) from two independent datasets (Human Connectome Project [HCP]; Leipzig Mind-Brain-Body dataset) (*Van Essen et al., 2013*; *Babayan et al., 2019*), along with a recently published multi-tracer receptor expression atlas derived from ~1000 positron emission tomography (PET) scans (*Hansen et al., 2022b*), we aim to provide insights on the spatial organization of connectivity within the human pulvinar and its relation to molecular expression.

## Results

### The multiscale gradient organization of the human pulvinar goes beyond discrete anatomical subdivisions

We collected functional and diffusion-weighted imaging data from two high-quality, independent datasets of healthy subjects: a primary dataset from the HCP (*Van Essen et al., 2012*), including 210 healthy subjects (males = 92, females = 118, age range 22–36 years), and a validation dataset from the Leipzig Mind-Brain-Body dataset (LEMON) (*Babayan et al., 2019*), consisting of 213 healthy subjects (males = 138, females = 75, age range 20–70 years). The pulvinar complex, along with its constituent subnuclei, was delineated bilaterally in the left and right hemispheres according to a whole-brain labeling atlas (*Rolls et al., 2020*). For each participant, we generated voxel-wise estimates of pulvinar functional and structural connectivity to a 400-label parcellation of the cerebral cortex (*Schaefer et al., 2018*), respectively, from preprocessed rs-fMRI and from whole-brain probabilistic tractograms derived from diffusion data. Individual connectivity maps were then normalized and aggregated to group-level, dense functional, and structural connectomes.

From the normative neurotransmitter atlas (*Hansen et al., 2022b*), we extracted the voxel-wise density profiles of 19 receptors and transporters across nine different neurotransmitter systems (serotonin, noradrenaline, dopamine, glutamate, gamma-aminobutyric acid, acetylcholine, histamine, opiates, endocannabinoids), and we calculated the pairwise correlation for each voxel (coexpression). The functional and structural dense connectomes, as well as the coexpression matrix, were converted to modality-specific affinity matrices quantifying the similarity of each voxel profile to each other.

By applying diffusion embedding (*Coifman and Lafon, 2006*) on modality-specific affinity matrices sampled from left and right pulvinar, we derived distinct functional connectivity, structural connectivity, and receptor expression embeddings (*Figure 1*).

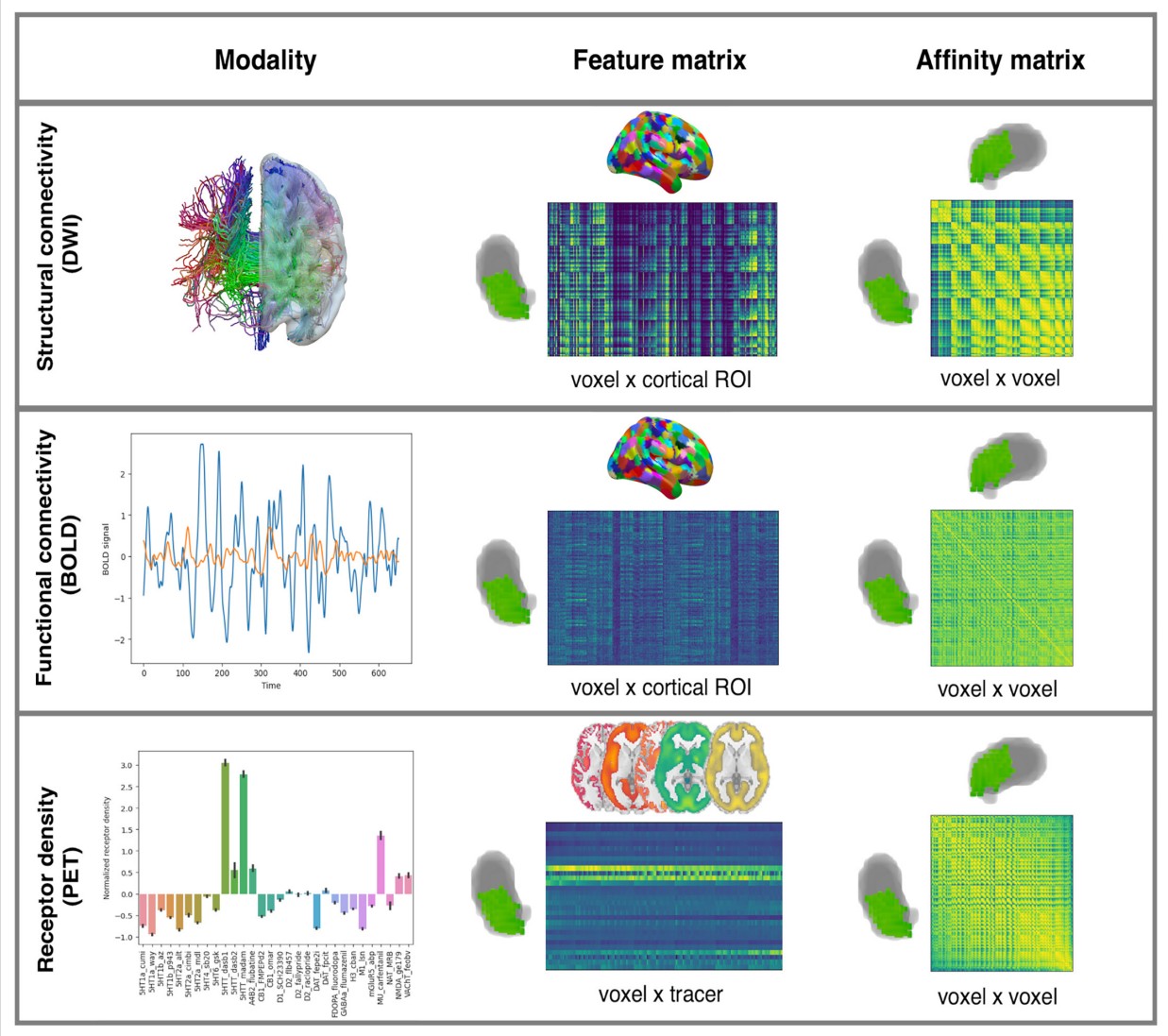

**Figure 1.** Schematic overview of the gradient mapping protocol. For each different imaging modality (DWI, BOLD, PET), a feature matrix was built by extracting voxel-wise features in the left and right pulvinar separately. Structural and functional connectivity to 400 cortical areas, as well as the expression profiles of 19 receptors and transporters, were extracted for each pulvinar voxel. Symmetric, affinity matrices comparing the feature profile similarity of each voxel against each other were estimated using cosine similarity and fed into the diffusion embedding algorithm. Additional details can be found in the main text. PET, positron emission tomography.

These embeddings, which we henceforth refer to as 'gradients', represent unitless entities in which each value identifies the position of pulvinar voxels along the respective embedding axis that encodes dominant differences in voxel connectivity or coexpression patterns (*Margulies et al., 2016*; *Haak et al., 2018*).

The first three functional connectivity gradients ($G_{FC}1$-$G_{FC}3$) collectively explained ~80% of the total variance in functional connectivity for both the left and right pulvinar, as evidenced by the elbow observed in the scree plot (*Figure 2A*). The principal gradient ($G_{FC}1$, ~50% of total variance explained for both left and right pulvinar) reflected a dorsomedial-to-ventrolateral topographical organization. Of the two secondary gradients, explaining each approximately ~10/15% of the total variance for both left and the right pulvinar, one ($G_{FC}2$) ran to the posteromedial part of the pulvinar to the most anterior dorsal region, the other ($G_{FC}3$) was aligned to the posterolateral to the anteromedial axis (*Figure 2A*).

Nearly 90% of the global variance in structural connectivity values was explained by the first two connectivity gradients ($G_{SC}1$-$G_{SC}2$). The first structural gradient ($G_{SC}1$, ~70% of total variance explained,

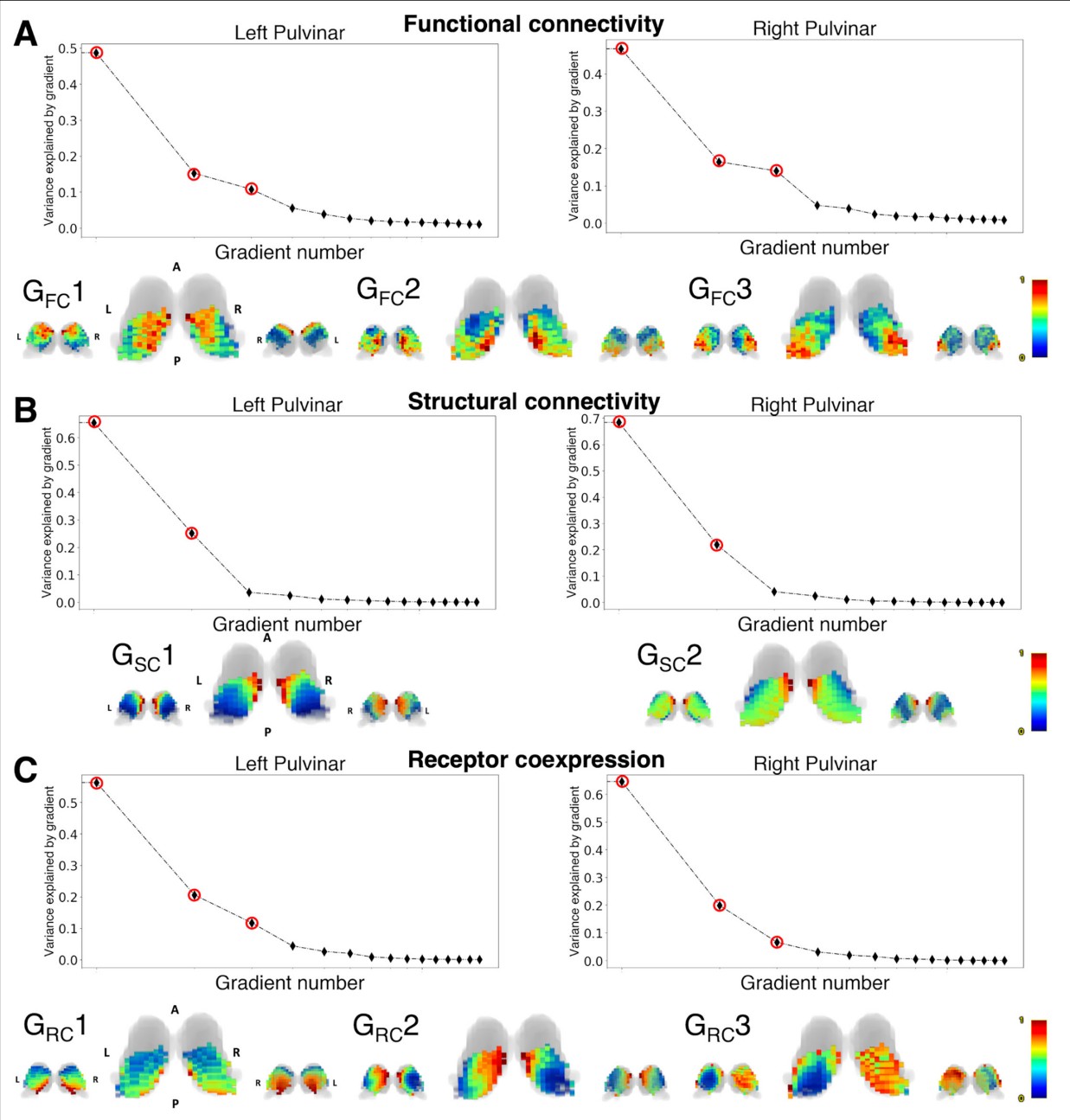

**Figure 2.** Functional connectivity (**A**), structural connectivity (**B**), and receptor coexpression (**C**) pulvinar gradients. For each panel, the top row shows the scree plots of explained variance for each gradient. The gradients that have been considered for subsequent analyses are marked with red circles. The bottom row shows 3D reconstruction of the normalized gradient images overlaid on the thalamic outline (gray-shaded area). The thalamic volume has been obtained from the AAL3 atlas. Gradients are shown in posterior (left), superior (center), and anterior (right) views. A: anterior; P: posterior; L: left; R: right.

The online version of this article includes the following figure supplement(s) for figure 2:

**Figure supplement 1.** Replication of functional (**A**), structural (**B**), and receptor coexpression (**C**) gradients on a thalamic-specific atlas (*Su et al., 2019*).

**Figure supplement 2.** K-means clustering of pulvinar gradients and their correspondence with histological pulvinar nuclei.

**Figure supplement 3.** Reliability and reproducibility of diffusion embedding of pulvinar-cortical connectivity data.

bilaterally) was arranged on a medio-lateral axis, while the second ($G_{SC}2$), explaining approximately 20% of global explained variance, delineated a dorsal-to-ventral topography (*Figure 2B*).

Finally, the first three gradient embeddings ($G_{RC}1$-$G_{RC}3$) explained globally ~80% of the variance in receptor coexpression data both for the left and right pulvinar, although with minor differences between the left and right hemispheres for individual gradients. The main gradient ($G_{RC}1$), explaining 60% of the total variance in the left hemisphere and 70% in the right, depicted a dorso-medial to ventro-lateral axis organization. The second gradient ($G_{RC}2$), accounting for ~20% of the explained variance on the left hemisphere and ~15% on the right hemisphere, displayed a medio-lateral topography. Finally, the third gradient ($G_{RC}3$), capturing approximately 10% of the total variance on the left and less than 10% on the right, ran along the anterior-posterior axis (*Figure 2C*).

Due to the small volume of the pulvinar complex, whose boundaries can vary significantly across different thalamic segmentation, we validated our results on an additional, thalamus-specific atlas (*Su et al., 2019*). We found that the results of the diffusion embedding are highly consistent, regardless of the imaging modality of choice, both in terms of variance explained and spatial topography (*Figure 2—figure supplement 1*).

To characterize the relationship between these pulvinar gradients and the conventional anatomical subdivision of the pulvinar complex into discrete nuclei, we analyzed the distribution of gradient values across the four traditional anatomical nuclei (medial, lateral, anterior, inferior) as defined by an atlas-based parcellation based on postmortem histology (*Iglesias et al., 2018*). First, by plotting gradient values grouped by each pulvinar nucleus, we observed limited correspondence to discrete pulvinar nuclei, regardless of the imaging modality. Although some gradients showed a progressing trend along different pulvinar nuclei, most gradient values were evenly distributed across nuclei. To assess if discrete pulvinar nuclear structure could be inferred from the continuous gradient values derived from diffusion embedding, k-means data-driven clustering in gradient space was applied, either by considering the most relevant gradients for each single imaging modality (functional, structural, or receptor coexpression) or by concatenating gradient values from all the modalities. The appropriate number of clusters (4 for functional gradients, 3 for structural gradients, 4 for coexpression gradients, and 8 for combined clustering) was selected based on silhouette score plots for each hemisphere. Clustering of gradient values resulted in generally poor overlap to anatomical nuclei either for single modality (maximum Dice coefficient left/right pulvinar: 0.45/0.42 to medial pulvinar for functional gradients, 0.67/0.66 to medial pulvinar for structural gradients; 0.61/0.46 to medial pulvinar for coexpression gradients) or concatenated modalities (maximum Dice coefficient left/right pulvinar: 0.67/0.57 to anterior pulvinar) (*Figure 2—figure supplement 2*).

## Pulvinar-cortical functional connectivity replicates patterns of cortico-cortical functional connectivity

Connectivity gradients serve as low-dimensional representations of the topographical organization patterns of connectivity, whether functional or structural connectivity. With this concept in mind, we sought to investigate the extent to which pulvinar-cortical connectivity reflects cortico-cortical connectivity. To achieve this, we generated pulvinar-cortical gradient-weighted connectivity maps for the most relevant structural and functional gradients. This involved calculating the average of the dot products of gradient values in each voxel and their connectivity to each cortical region of interest (ROI). Subsequently, we correlated these maps with gradients derived from the embedding of cortico-cortical connectivity. To address the issue of spatial autocorrelation (SA), which commonly affects brain maps, we computed SA-corrected permutational p-values using a method based on SA-preserving surrogates (*Burt et al., 2020*). Furthermore, these SA-corrected p-values were adjusted for false discovery rate (FDR) using the Benjamini-Hochberg correction method.

Diffusion embedding of cortico-cortical functional connectivity gradients revealed an elbow in the explained variance graph at the fifth gradient, collectively explaining ~55% of the total variance (*Figure 3—figure supplement 1*). Similarly, the first four gradients of cortico-cortical structural connectivity accounted for ~60% of the total variance (*Figure 3—figure supplement 2*).

We found that pulvinar-cortical connectivity patterns associated with pulvinar gradients exhibited high correlations with the first three cortico-cortical functional connectivity gradients. Specifically, $G_{FC}1$-weighted connectivity maps demonstrated the highest correlation with the first cortico-cortical functional gradient (left pulvinar: r=0.89, p<0.01; right pulvinar: r=0.83, p<0.01); in line with the

available evidence, this gradient delineates a unimodal-to-transmodal cortical hierarchy, spanning from low-level sensory and motor regions to higher-order associative and limbic cortices (*Margulies et al., 2016*). $G_{FC}2$-weighted connectivity maps strongly correlated with the third cortico-cortical functional gradient (left pulvinar: r=0.71, p<0.01; right pulvinar: r=0.67, p<0.01); as in previous investigations, the cortical topography of this gradient spans across cortical regions involved in low-versus-high complexity cognitive task, thus highlighting a cortical organization depending on cognitive demand (*Turnbull et al., 2020*). Finally, $G_{FC}3$-weighted connectivity was found strongly correlated with the second cortico-cortical functional gradient (left pulvinar: r=0.78, p<0.01; right pulvinar: r=0.78, p<0.01); in line with previous works, this cortical gradient is anchored on one end on visual processing areas, and on the opposite end on sensorimotor processing networks, reflecting a secondary cortico-cortical hierarchy that is orthogonal to the principal cortical gradient (*Margulies et al., 2016*). Additionally, weaker yet significant correlations were found between $G_{FC}2$-weighted connectivity maps and the first cortico-cortical functional gradient (left pulvinar: r=0.27, p=0.01; right pulvinar: r=0.38, p<0.01) (*Figure 3A*).

On the contrary, gradient-weighted structural connectivity maps did not exhibit the same correlation pattern with cortico-cortical structural gradient maps, displaying only weak and not significant correlations to cortical gradients. Notably, left $G_{SC}2$-weighted structural connectivity maps showed weak correlations with the principal structural connectivity gradient (r=0.19, p>0.01), whereas right $G_{SC}2$-weighted connectivity weakly correlated with the second cortico-cortical gradient (r=0.24, p>0.01) (*Figure 3—figure supplement 2*).

## The unimodal-to-transmodal gradient ($G_{FC}1$) aligns with receptor expression on the dorso-ventral pulvinar axis

We discovered that the main gradient of cortico-cortical functional connectivity is mirrored on the pulvinar by $G_{FC}1$. The cortical connectivity of this gradient progresses from unimodal cortical areas of the sensorimotor and visual network to multimodal, higher-order associative areas of the limbic, frontoparietal, and default mode network (*Figure 3B*). Similarly, in the pulvinar, this trend manifests as a progression of gradient values in a ventral-to-dorsal fashion, from the anterior and inferior to the lateral and medial pulvinar nuclei (*Figure 4B*).

We further delved into the relationship between multi-scale organization features of the pulvinar by correlating gradients obtained from embedding data obtained from different imaging modalities. To take into account the effects of SA, we corrected the resulting p-values using a method based on SA-preserving spatial null models (*Burt et al., 2020*). For this gradient, we found the strongest association with the main gradient of receptor expression $G_{RC}1$ (*Figure 4A*; left pulvinar: r=−0.71, p<0.01; right pulvinar: r=−0.78, p<0.01). It is worth noting that, due to the intrinsic sign indeterminacy of gradient values, absolute correlation values were considered. Similarly to $G_{FC}1$, $G_{RC}1$ was also organized along the dorso-ventral axis of the pulvinar complex and displayed a similar distribution of values across pulvinar nuclei, albeit inverted with respect to $G_{FC}1$, with loadings progressing from lateral and medial to anterior and inferior pulvinar nuclei.

To identify the receptor coexpression patterns driving this specific mode of organization, we examined the values of the five tracers with the strongest correlation to gradient values. We observed that expression of the serotonin and noradrenaline reuptake transporters (5HTT and NAT) positively correlated with gradient values. Conversely, the expression of markers of dopaminergic activity, such as the D2 receptor or the reuptake transporter DAT, as well as NMDA glutamatergic receptors, decreased with increasing gradient values (*Figure 4C*).

Furthermore, we observed a significant correlation between $G_{FC}1$ and the secondary gradient of structural connectivity $G_{SC}2$ (left pulvinar: r=0.70, p<0.01; right pulvinar: r=0.72, p<0.01), which explained ~20% of the variance in structural connectivity. Similar to $G_{FC}1$, $G_{SC}2$ follows the dorso-ventral axis of the pulvinar complex, progressing from lower values in the anterior and lateral nuclei to higher values in the inferior and medial nuclei (*Figure 4C*). Consequently, this gradient also displayed a weaker negative correlation with $G_{RC}1$ (left pulvinar: r=−0.44, p<0.01; right pulvinar: r=−0.46, p<0.01). However, the cortical structural connectivity pattern associated with $G_{FC}1$ does not mirror that of its functional connectivity counterpart. Instead, gradient-weighted structural connectivity progresses from lower values in modality-independent networks (e.g. default mode, frontoparietal) to intermediate values in unimodal, low-level sensory networks (e.g. visual and sensorimotor) with

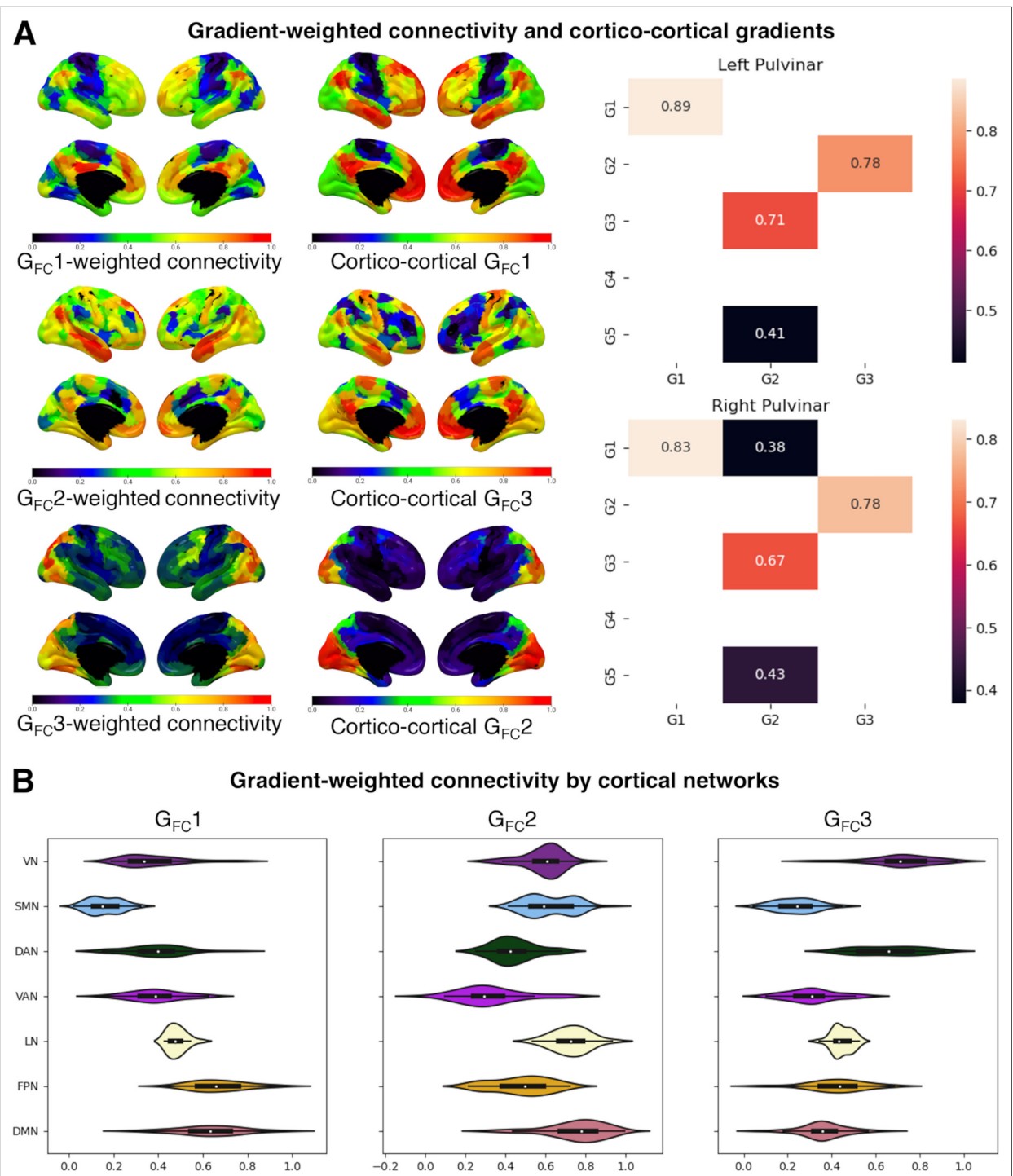

**Figure 3.** Gradient-weighted connectivity and cortico-cortical gradients. (**A**) The relationship between pulvinar-cortical and cortico-cortical functional gradients. The left panels show gradient-weighted connectivity values for the first three cortico-pulvinar functional connectivity gradients, each paired with the most correlated cortico-cortical functional connectivity gradient. Normalized values of each cortical region of interest (ROI) (Schaefer atlas, 400 parcels) are plotted on the cortical surface. The right panels show matrix plots of Pearson's correlation values between cortico-cortical gradients (y-axis) and pulvinar-cortical gradient-weighted connectivity maps. Only values showing statistical significance (spatial autocorrelation [SA]-corrected, false discovery rate [FDR]-adjusted p<0.01) are shown. (**B**) The relationship between pulvinar-cortical functional gradients and cortical connectivity networks. Violin plots of normalized gradient-weighted connectivity values grouped by seven intrinsic connectivity networks (as in **Thomas Yeo et al., 2011**); VN: visual network; SMN: sensorimotor network; DAN: dorsal attention network; VAN: ventral attention network; LN: limbic network; FPN: frontoparietal network, DMN: default-mode network.

*Figure 3 continued on next page*

*Figure 3 continued*

The online version of this article includes the following figure supplement(s) for figure 3:

**Figure supplement 1.** Cortico-cortical functional connectivity gradients.

**Figure supplement 2.** Cortico-cortical structural connectivity gradients and their relationship to pulvinar-cortical structural gradients.

maximal values in networks involved in higher-order sensory processing (e.g. limbic, ventral attention, dorsal attention).

In summary, our findings demonstrated that $G_{FC}1$, $G_{RC}1$, and $G_{SC}2$ substantially delineate multiscale differences between the ventral and dorsal aspects of the pulvinar. Moving along the ventral-dorsal axis of the pulvinar complex, more ventral regions showed higher functional connectivity to unimodal sensory processing networks, higher levels of 5HTT and NAT expression, and preferentially higher structural connectivity to modality-independent or low-level sensory processing cortices. Conversely, dorsal regions displayed increasingly higher functional connectivity to multimodal processing networks, higher levels of D2, NMDA receptors, and greater structural connectivity to higher-order sensory processing cortices (*Figure 4*).

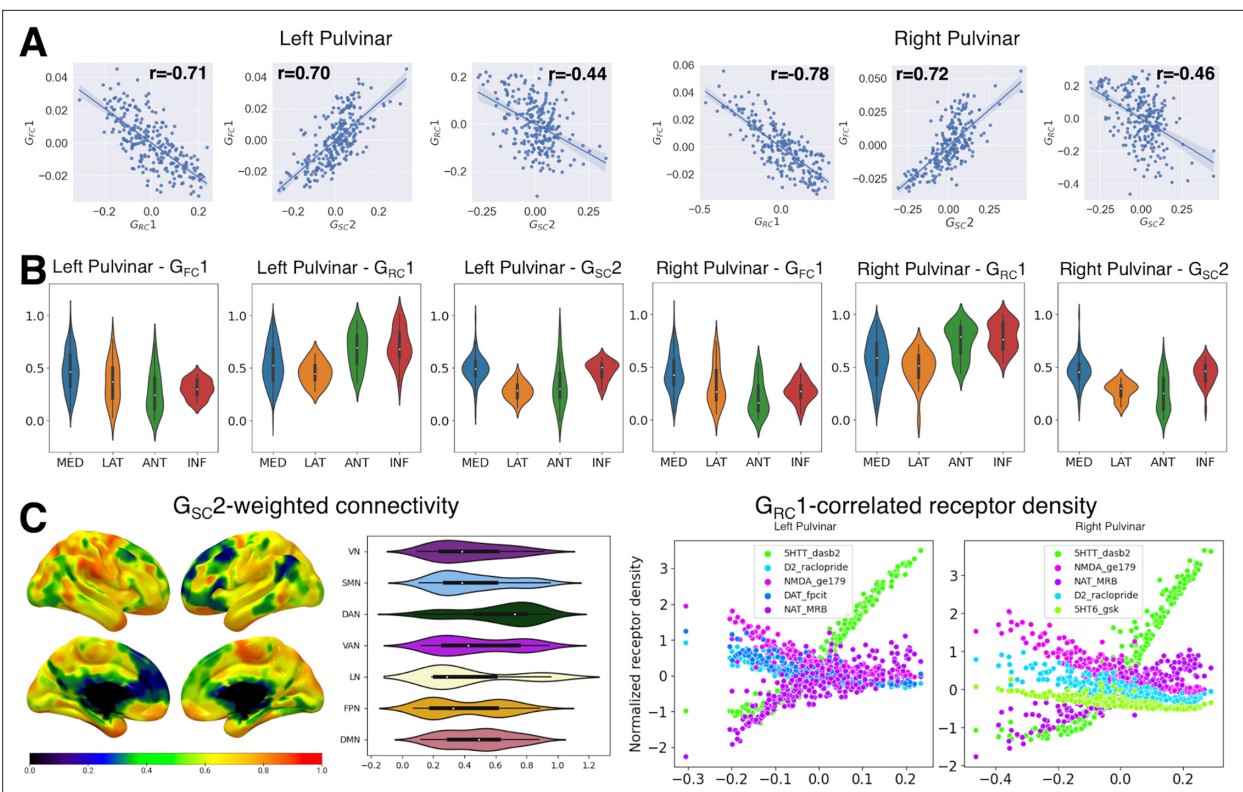

**Figure 4.** Pulvinar correlates of the unimodal-transmodal cortical gradient. (**A**) Scatter plots illustrate the relationship between the first pulvinar-cortical connectivity gradient, corresponding to the unimodal-transmodal hierarchy of cortico-cortical connectivity, and its most correlated gradient values across other modalities. Due to the intrinsic sign indeterminacy of gradient values, absolute correlation values are considered. (**B**) The relationship between correlated pulvinar gradients and discrete histological nuclei. Violin plots illustrating normalized gradient values grouped by histological nuclei (AAL atlas). MED: medial pulvinar; LAT: lateral pulvinar; ANT: anterior pulvinar; INF: inferior pulvinar. (**C**) Structural connectivity and receptor coexpression patterns correlating with the unimodal-transmodal hierarchy. Left panels: gradient-weighted structural connectivity. Normalized values for each cortical region of interest (ROI) (Schaefer atlas, 400 parcels) are plotted on the cortical surface. Violin plots show values grouped by seven intrinsic connectivity networks (as in *Thomas Yeo et al., 2011*). VN: visual network; SMN: sensorimotor network; DAN: dorsal attention network; VAN: ventral attention network; LN: limbic network; FPN: frontoparietal network; DMN: default-mode network. Right panels: gradient-correlated receptor density values for the top 5 most correlated receptors. Details can be found in the main text.

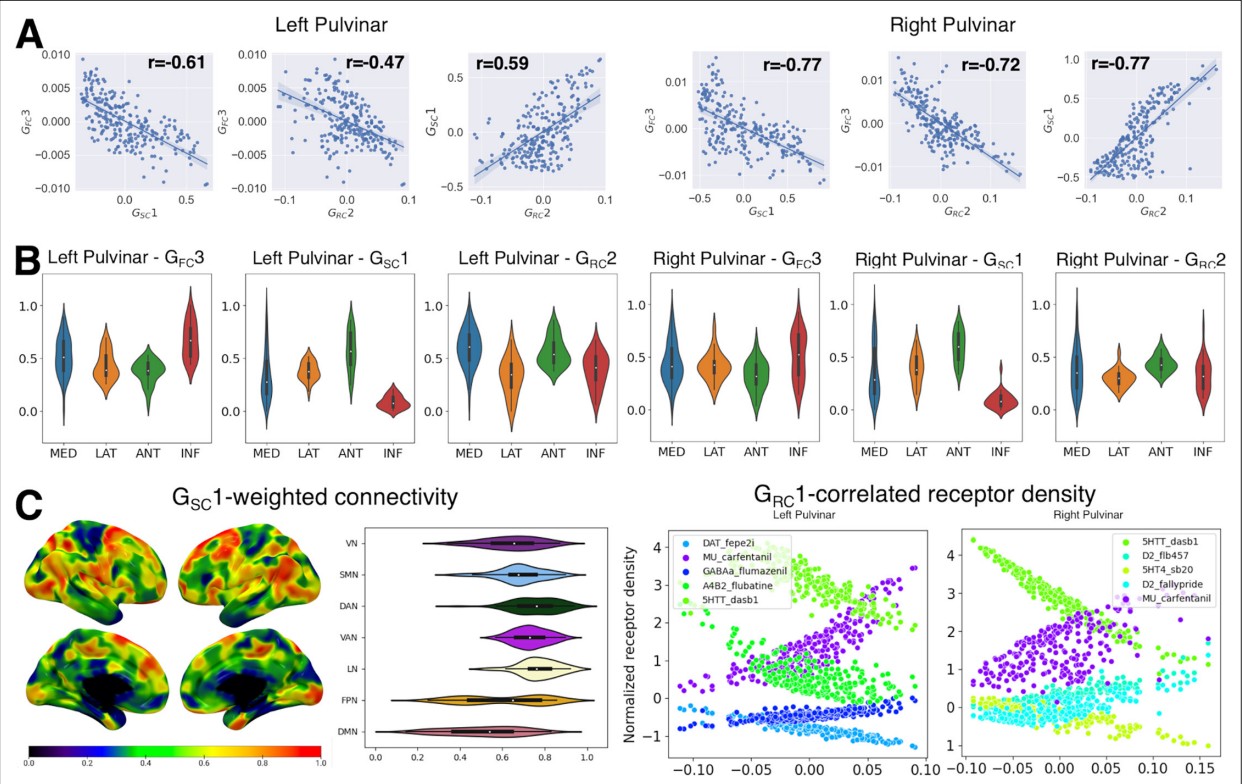

**Figure 5.** Pulvinar correlates of the visual-to-sensorimotor cortical gradient. (**A**) Scatter plots illustrate the relationship between the third pulvinar-cortical connectivity gradient, corresponding to the visual-to-sensorimotor hierarchy of cortico-cortical connectivity, and its most correlated gradient values across other modalities. Due to the intrinsic sign indeterminacy of gradient values, absolute correlation values are considered. (**B**) The relationship between correlated pulvinar gradients and discrete histological nuclei. Violin plots illustrating normalized gradient values grouped by histological nuclei (AAL atlas). MED: medial pulvinar; LAT: lateral pulvinar; ANT: anterior pulvinar; INF: inferior pulvinar. (**C**) Structural connectivity and receptor coexpression patterns correlating with the visual-to-sensorimotor hierarchy. Left panels: gradient-weighted structural connectivity. Normalized values for each cortical region of interest (ROI) (Schaefer atlas, 400 parcels) are plotted on the cortical surface. Violin plots show values grouped by seven intrinsic connectivity networks (as in ***Thomas Yeo et al., 2011***). VN: visual network; SMN: sensorimotor network; DAN: dorsal attention network; VAN: ventral attention network; LN: limbic network; FPN: frontoparietal network, DMN: default-mode network. Right panels: gradient-correlated receptor density values for the top 5 most correlated receptors. Details can be found in the main text.

## The visual-to-sensorimotor gradient ($G_{FC}3$) aligns with structural connectivity on the medio-lateral pulvinar axis

The secondary gradient of cortico-cortical organization was found to be correlated with $G_{FC}3$, which explained ~10% of the variance in pulvinar-cortical functional connectivity. $G_{FC}3$ exhibited a well-delineated medial-lateral and antero-posterior arrangement, corresponding to lower values in the anterior and lateral pulvinar divisions and higher values in the inferior and medial nuclei. The cortical functional connectivity patterns associated with $G_{FC}3$ spanned from the lower extreme of the sensorimotor network to the higher extreme of the visual network. This gradient reflected transitions from increasingly visual-related processing networks (such as dorsal attention and limbic networks) to increasingly action-related processing networks (including the default mode, frontoparietal, and ventral attention networks) (***Figure 5B***).

We found that $G_{FC}3$ is the most correlated, in absolute values, to the principal gradient $G_{SC}1$, which explained ~70% of variance in pulvinar-cortical structural connectivity, showing a strong negative correlation (***Figure 5A***; left pulvinar: r=–0.61, p<0.01; right pulvinar: r=–0.77, p<0.01). $G_{SC}1$ also delineates a clear progression along the medio-lateral axis of the pulvinar, with lower values in the inferior and medial pulvinar nuclei transitioning to higher values in the lateral and anterior pulvinar nuclei (***Figure 5B***). On the cortical mantle, the structural gradient-weighted connectivity patterns exhibited a heterogeneous distribution of values within functional networks, rather than delineating a specific pattern of progression across functional networks (***Figure 5C***).

Additionally, we found a significant negative correlation to $G_{RC}2$, which accounts for ~15% of the variance in receptor expression (left pulvinar: r=−0.47, p<0.01; right pulvinar: r=−0.72, p<0.01). This gradient consistently showed an association in both hemispheres with the expression of mu-opioid receptors (MOR), which was positively correlated to the gradient axis, and of the 5HTT, which was negatively correlated to the gradient axis (*Figure 5C*). Similar to $G_{FC}3$, $G_{RC}2$ was also found to be correlated with the main gradient of structural connectivity $G_{SC}1$ (*Figure 5A*; left pulvinar: r=0.59, p<0.01; right pulvinar: r=0.77, p<0.01). Like $G_{SC}1$, $G_{RC}2$ exhibited a progression of values from inferior and medial pulvinar nuclei to the lateral and anterior nuclei (*Figure 5B*).

To summarize, our findings reveal a strong association between the medio-lateral axis of pulvinar organization, which reflects the progression of functional connectivity from sensorimotor to associative and visual areas, and the principal gradient of structural connectivity. The cortical connectivity profiles of this gradient delineate within-network patterns of progression, suggesting that multiple functional networks host separate representations of the pulvinar complex. Additionally, this structural and functional connectivity pattern is paralleled by a secondary gradient of receptor expression. Specifically, there is an increasing expression of MOR and a decreasing expression of 5HTT in the medial end of the gradient, which corresponds to regions functionally connected to sensorimotor-associative regions. Conversely, there is a decreasing expression of MOR and an increasing expression of 5HTT in the lateral end of the gradient, which corresponds to regions functionally connected to associative-visual regions (*Figure 5*).

## Reliability and reproducibility

Diffusion embedding of cortico-pulvinar connectivity data demonstrated high stability and reproducibility, with structural connectivity gradients exhibiting superior performance compared to functional connectivity gradients.

The structural connectivity gradients $G_{SC}1$ and $G_{SC}2$ showed excellent stability across 100 resampling of random halves of the main dataset in both hemispheres (all median r>0.99 for both left and right pulvinar, with IQR <0.01). Additionally, group-level structural gradients demonstrated the highest repeatability when compared between test and retest samples of the main dataset (all r>0.99 for both left and right pulvinar). Finally, structural gradients showed high reproducibility across both the main and validation datasets (left pulvinar, $G_{SC}1$: r=0.87; $G_{SC}2$: r=0.88; right pulvinar, $G_{SC}1$: r=0.87; $G_{SC}2$: r=0.85).

Functional connectivity gradients showed overall very high or moderate-to-high stability across split-half resamples of the main dataset in both hemispheres. However, stability decreased with decreasing variance explained (left pulvinar, $G_{FC}1$: median r=0.85, IQR = 0.021; $G_{FC}2$: median r=0.70, IQR = 0.042; $G_{FC}3$: median r=0.59, IQR = 0.061; right pulvinar, $G_{FC}1$: median r=0.87, IQR = 0.023; $G_{FC}2$: median r=0.75, IQR = 0.039; $G_{FC}3$: median r=0.72, IQR = 0.039).

Repeatability on test-retest samples of the main dataset also showed a decreasing trend following the decrease of explained variance, with moderate to good repeatability for $G_{FC}1$ and $G_{FC}2$, and low repeatability for $G_{FC}3$. In contrast, reproducibility of the group-level gradients across the main and validation sample, while showing a similar trend with higher values for $G_{FC}1$ compared to the other secondary gradients, was found high both for left (all r>0.75) and right pulvinar (all r>0.80).

To assess the influence of local noise on functional and structural connectivity gradients, we calculated the spatial correlation between gradient values and averaged voxel-wise estimates of signal-to-noise ratio (SNR) from fMRI and structural MRI data, respectively. We found that functional connectivity gradients are weakly, but significantly correlated with the SNR, with the strongest correlation observed for the third gradient (left hemisphere $G_{FC}1$ r=−0.30, SA-corrected p<0.001, $G_{FC}2$ r=0.22, SA-corrected p=0.05, $G_{FC}3$ r=0.55, SA-corrected p<0.001; right hemisphere $G_{FC}1$ r=−0.41, SA-corrected p<0.001, $G_{FC}2$ r=0.22, SA-corrected p=0.008, $G_{FC}3$ r=0.52, SA-corrected p=0.017). In contrast, structural connectivity gradients were not significantly associated with SNR (left hemisphere $G_{SC}1$ r=0.06, SA-corrected p=0.82, $G_{SC}2$ r=−0.33, SA-corrected p=0.01; right hemisphere $G_{SC}1$ r=0.40, SA-corrected p=0.28, $G_{SC}2$ r=−0.19, SA-corrected p=0.31) (*Figure 2—figure supplement 3*).

## Discussion

In the present study, we employed a data-driven dimensionality reduction analysis on a comprehensive array of high-resolution, large-scale multimodal datasets encompassing structural, functional connectivity, and receptor expression data. Our objective was to delve into the organizational principles underlying pulvinar-cortical connectivity and its relationship to cortico-cortical connectivity. Through the application of diffusion embedding, a widely utilized gradient analysis technique, we successfully collapsed the high-dimensional connectivity and coexpression data into a reduced set of low-level representations, reflecting the spatial patterns of organization of multimodal features on the pulvinar complex.

Previous reports on functional or structural connectivity of the human pulvinar complex have been predominantly either ROI-based and hypothesis-driven (*Leh et al., 2008*; *Tamietto et al., 2012*; *Arcaro et al., 2015*; *Basile et al., 2021*) and thus unable to capture finer details regarding the topographical organization patterns of pulvinar connectivity in its entirety, or data-driven and focused on hard parcellations from data clustering algorithms (*Barron et al., 2015*; *Guedj and Vuilleumier, 2020*). The later methodological approach tends to force the topographical heterogeneity of pulvinar connectivity patterns into discrete, spherical units rather than providing insights into their continuous transitions in space.

Over the past five decades, the connectivity of the pulvinar complex has undergone investigations in nonhuman primates, utilizing anatomical tract-tracing or electrophysiological methods (*Asanuma et al., 1985*; *Baleydier and Mauguiere, 1985*; *Pons and Kaas, 1985*; *Cusick and Gould, 1990*; *Baleydier and Morel, 1992*; *Romanski et al., 1997*; *Shipp, 2001*). A prevailing principle suggests that connectivity to cortical regions does not adhere to clear-cut distinctions between histological subdivisions of the pulvinar complex but instead follows a topographical organization spanning across anatomical nuclei (*Shipp, 2003*). Accordingly, existing connectivity-based parcellation models of the pulvinar complex have generally shown moderate-to-low correspondence between functional connectivity or coactivation clusters and histological delineations of pulvinar nuclei (*Barron et al., 2015*; *Ji et al., 2016*; *Guedj and Vuilleumier, 2020*). Our results align with this perspective by demonstrating that while structural and functional connectional topography often follows recognizable patterns of progression across anatomical nuclei, hard clustering of either connectivity or coexpression data failed to reveal the underlying boundaries of nuclear anatomy. In other words, the topographical pattern of connection and coexpression only partially reflects the anatomical boundaries outlined by the connectivity or coexpression differences between nuclei.

It is worth noting, however, that connectivity profiles of individual pulvinar nuclei can be properly inferred from gradient values and their relative gradient-weighted connectivity maps. For instance, $G_{FC}1$, which showed progression of gradient-weighted connectivity from low-level sensorimotor and visual processing regions to higher-level multimodal, limbic, and default-mode regions, demonstrated a corresponding progression of values from anterior (mostly sensorimotor) and inferior (low-level visual) nuclei to lateral (higher-level visual) and medial (associative and limbic) pulvinar nuclei. This observation is consistent with existing literature in nonhuman primates (*Benarroch, 2015*; *Homman-Ludiye and Bourne, 2019*). In synthesis, the notion of connectivity gradients reconciles the concept of continuous and graded cortical representations on the pulvinar complex with the hypothesis of functional and anatomical specialization of discrete pulvinar subregions.

Drawing from anatomical literature in primates, *Shipp, 2003*, proposed a model of pulvinar-cortical connectivity, positing a clear dissociation between dorsal and ventral pulvinar, with each region hosting distinct and parallel representations of cortical regions along their latero-medial axis. According to this model, connections to the dorsal pulvinar are organized along a dorsal parietal-superior temporal axis, while connections to the ventral pulvinar follow an occipital-inferior temporal axis. Subsequent investigations using rs-fMRI and diffusion tractography in the human brain have further substantiated this specific organizational pattern (*Arcaro et al., 2015*; *Arcaro et al., 2018*). Findings from connectivity-based parcellation studies substantially agree with this model. Anterior (and ventral) clusters predominantly exhibit connections to precentral and postcentral gyri; dorsomedial clusters show connectivity to the cingulum, precuneus, and inferior parietal lobules; dorsal-lateral clusters connected to prefrontal and parietal regions and a ventromedial cluster displaying connectivity to sensorimotor and occipitotemporal cortical regions (*Guedj and Vuilleumier, 2020*; *Basile et al., 2024*).

Consistent with these findings, our study reveals that the two major axes of both structural and functional connectivity converge on the dorso-medial and ventro-lateral axes of the pulvinar complex, confirming the double dissociation pattern described therein. Ventral regions of the pulvinar (low on $G_{FC}1$) exhibited stronger connections to occipital and inferior temporal regions, while dorsal regions (high on $G_{FC}1$) were predominantly connected to frontoparietal and cingulate regions. Consequently, the secondary gradient of pulvinar-cortical functional connectivity progressed from lower extreme values in the dorsolateral pulvinar (associated with connectivity to frontoparietal regions) to dorso-medial pulvinar (mostly connected to superior temporal cortex), passing through ventromedial and ventrolateral pulvinar (respectively connected to occipital and inferior temporal lobes). Lastly, medial regions of the pulvinar (low on $G_{FC}3$) exhibited stronger connections to the occipital and parietal lobe compared to lateral regions (high on $G_{FC}3$), which were instead more correlated with superior and inferior temporal cortices.

Our findings provide a nuanced perspective on Shipp's model of pulvinar-cortical connectivity, shedding light on the intricate relationships between pulvinar-cortical and cortico-cortical connection patterns. While anatomical studies in nonhuman primates have suggested that highly connected cortical areas may share overlapping representations on the pulvinar complex, known as the 'replication principle' (*Shipp, 2003*), our work offers robust formal evidence of this organizational principle in the human brain. Although we could not replicate the same results for structural connectivity, likely due to the inherent disproportion between pulvinar-cortical connections and the whole-brain connectome, we demonstrated a one-to-one correspondence between the first three cortico-cortical functional connectivity gradients and the three main functional connectivity gradients of the pulvinar.

Beyond merely confirming that cortical functional networks converge on shared connectional domains within the pulvinar complex, our findings underscore how pulvinar-cortical connectivity closely mirrors the organizational principles observed in the cerebral cortex. These principles, initially elucidated in the seminal work of *Margulies et al., 2016*, and grounded in earlier anatomical models based on tract-tracing studies in nonhuman primates (*Mesulam, 1998*), reflect a hierarchical patterning of information transfer from unimodal sensory or motor representation toward increasing levels of integration and abstraction in multimodal associative cortices.

Specifically, the principal gradient of cortical connectivity, extending from primary visual, somatosensory, motor, and auditory cortices to associative regions of the limbic, frontoparietal, and default-mode network, not only accounted for the largest amount of variance of pulvinar functional connectivity, but also delineated the ventro-dorsal pulvinar axis. Likewise, the gradient capturing the second-most cortical connectivity variance, which spans from visual to motor-auditory unimodal cortices with modality-independent regions situated at various levels, corresponds to the third pulvinar-cortical gradient in terms of explained variance, mapped along the medio-lateral axis. Interestingly, the third gradient of cortico-cortical connectivity, in terms of pulvinar-cortical connectivity, appears slightly more represented than this secondary, cross-modal gradient, as indicated by the higher explained variance. This cortical gradient, extending from dorso-lateral to dorso-medial pulvinar regions, progresses from task-negative to task-positive networks of the cerebral cortex and has been linked to spontaneous attention during naturalistic tasks (*Samara et al., 2023*).

Together, these results reinforce the concept of pulvinar involvement in multimodal sensory integration and top-down attentional processing, consistent with recent literature (*Saalmann et al., 2012*; *Benarroch, 2015*; *Fiebelkorn and Kastner, 2020*; *Froesel et al., 2021*). They also align with the proposed role of pulvinar-cortical connectivity in enhancing communication and synchronization within cortical processing modules in a context-dependent and competitive manner (*Rockland et al., 1999*; *Cortes et al., 2021*; *Cortes et al., 2024*; *Aussel et al., 2023*; *Bastuji et al., 2024*).

In this context, it could be hypothesized that the observed gradient organization of the pulvinar may also exhibit specific patterns in the temporal domain. Indeed, multiple investigations have linked the temporal dynamics of cortical regions to different aspects of information processing. Notably, intrinsic neural timescales of functional activity have been associated with the functional specialization and gradient organization of the cerebral cortex (*Golesorkhi et al., 2021*), with shorter timescales in unimodal sensory regions and longer ones in transmodal networks (*Ito et al., 2020*; *Murray et al., 2014*). Moreover, thalamocortical connectivity has been shown to correlate with these patterns of intrinsic timescale (*Müller et al., 2020*). In addition, modulatory neurotransmitters such as serotonin and dopamine have been demonstrated to play a significant role in modulating functional cortical

dynamics across different timescales (*Hansen et al., 2022b*; *Luppi et al., 2023*). Exploring how the spatial organization of the pulvinar relates to temporal dynamics and timescale modulation could provide valuable insights and represents a promising avenue for future investigations.

In recent years, different works have explored the spatial arrangement of thalamic connectivity within a connectivity gradient framework. Diffusion embedding of thalamocortical functional connectivity has revealed a principal, medio-lateral gradient that was found correlated with thalamic structural subdivisions, and a secondary, antero-posterior gradient associated with thalamic functional subfields, showing progression from unimodal sensorimotor cortical networks to multimodal attention and associative networks. Interestingly, the principal thalamic gradient shows a medio-lateral arrangement on the pulvinar axis while the secondary gradients correspond more to a ventral-dorsal pulvinar axis (*Yang et al., 2020*). In particular, further independent investigations have suggested that the progressing pattern of thalamic connectivity from unimodal to transmodal cortices is strongly associated with the local density of core and matrix cell types, thus establishing a link between molecular properties and functional connectivity dynamics (*Müller et al., 2020*; *Huang et al., 2024*). Our findings complement and expand the existing literature by revealing a similar arrangement of cortical connectivity patterns on the pulvinar complex, elucidating its relationship to in vivo estimates of molecular markers of neurotransmission. We found that the gradient associated with unimodal-transmodal cortical connectivity accounted for the highest percentage of variance in cortico-pulvinar connectivity, in line with its well-acknowledged role of associative nucleus. It is noteworthy that, in analyses of thalamocortical gradients, the pulvinar complex is situated toward the 'sensorimotor' extreme of the unimodal-to-transmodal thalamic gradient (*Yang et al., 2020*). This likely reflects its prominent connectivity to visual and sensory areas compared to other thalamic nuclei. Nevertheless, the extensive and intricate association of pulvinar with multiple cortical networks is strongly evident in various functional connectivity investigations (*Basile et al., 2021*; *Kumar et al., 2017*, *Kumar et al., 2022*). By isolating pulvinar-cortical from broader thalamocortical connectivity, our analysis was able to provide additional insights into the spatial organization of its connectivity with different cortical networks, highlighting the pulvinar's remarkable functional diversity and complexity.

As regards structural connectivity, existing accounts describe a medio-lateral organization of thalamocortical connections, corresponding to an antero-posterior gradient on the cortical mantle. This gradient organization appears to be anchored to genetic markers of different cell types (*Oldham and Ball, 2023*). In line with their findings, we describe a principal axis of structural connectivity in the pulvinar complex that is arranged on the medio-lateral axis, and we enforce the notion of a deep relationship between structural connections and molecular expression of neurotransmission markers. On the other hand, the patterns of connectivity with the cerebral cortex do not correspond to a clear antero-posterior axis on the cerebral cortex, probably showing the predominance of local connectivity over the global thalamic structural topography. Further investigations are warranted to ascertain whether the structural gradients of the pulvinar complex may be in continuity with this general cortico-thalamic connectivity gradient.

## Neurochemical correlates of pulvinar-cortical topographical organization

In addition to elucidating the intricate pulvinar-cortical connectivity patterns, our study unveils unprecedented insights into the correlates of this complex topographical organization at multiple organizational scales, including structural connectivity and neurotransmission. Notably, we found that the hierarchical organization of pulvinar-cortical connectivity aligns with the principal axis of receptor expression, indicating a substantial influence of neuromodulator receptor expression on this level of organization. This principal axis of receptor expression reflects diverging expression patterns of markers of catecholaminergic neurotransmission, encompassing dopaminergic, noradrenergic, and serotonergic systems. While previous ex vivo and in vivo investigations in the human and nonhuman primate brain confirm that pulvinar is a recipient of dopaminergic, noradrenergic, and serotonergic projections (*Lavoie and Parent, 1991*; *Oke et al., 1997*; *Rieck et al., 2004*; *Sánchez-González et al., 2005*; *Pérez-Santos et al., 2021*), our work provides information on their specific distribution pattern, representing a novel contribution to the field. Moreover, studies in nonhuman primates have highlighted many receptor markers whose topographic distribution mirrors connectivity patterns in the pulvinar complex, including acetylcholine esterase (AChE), parvalbumin, calbindin, or vesicular

glutamate transporters 1 and 2 (*Gutierrez et al., 1995*; *Gutierrez et al., 2000*; *Stepniewska and Kaas, 1997*; *Balaram et al., 2013*; *Balaram et al., 2015*). Since some of these markers are known to colocalize with catecholaminergic receptors in different brain areas (*Liang et al., 1996*; *Graham et al., 2015*) and have been recently correlated with functional topography in the cortex and subcortex, including the thalamus (*Anderson et al., 2018*; *Anderson et al., 2020*; *Müller et al., 2020*), the macroscale pattern of receptor coexpression observed by PET may reflect broader chemoarchitectural features at the cellular level.

Indeed, while commonalities and discrepancies between structural and functional connectivity have been extensively investigated, the relationship between functional connectivity and modulatory neurotransmission remains poorly understood. Our findings reveal stronger associations between pulvinar-cortical connectivity to specific functional networks and the spatial distribution of markers of serotonergic, noradrenergic, dopaminergic, and opioid systems. Pharmacological challenge studies using rs-fMRI suggest that each of these neurotransmission systems may either directly modulate thalamocortical connectivity (*Salomon et al., 2012*; *Cole et al., 2013a*; *Cole et al., 2013b*; *Farr et al., 2014*; *Lewis et al., 2021*; *Pizzi et al., 2023*) or influence neuronal gain in cortico-cortical functional connectivity (*Shine et al., 2018*), which is known to depend, in part, on cortical connections to associative thalamic nuclei, including the pulvinar.

Indeed, serotonergic neurotransmission in the thalamus plays a crucial role in higher-order sensory integration, especially within the visual system. In this regard, alterations in thalamo-cortical functional connectivity, influenced by powerful serotonergic psychedelics, are known to mediate this process (*Preller et al., 2018*; *Gaddis et al., 2022*; *Onofrj et al., 2023*). This perspective sheds light on the increasing 5HTT expression from high- to low-level somatosensory and visual regions of the pulvinar. Conversely, the increasing expression of dopamine receptors and transporters observed in the dorsal, higher-order regions of the pulvinar is consistent with the role of dopamine neuromodulation in mediating attentional processes, as supported by the findings of increased functional connectivity between thalamus and higher-order cortical regions after administration of the selective DAT-blocker modafinil (*Pizzi et al., 2023*). Finally, recent multimodal investigations have reported higher levels of dopamine receptors and transporters in cortical and subcortical regions structurally and functionally connected with the dorsal attention network, including the dorsomedial pulvinar (*Alves et al., 2022*). This further supports our results and underscores the pivotal role of dopamine neuromodulation in attentional processes.

## A multimodal pulvinar gradient architecture as a possible benchmark for neurological disorders

By integrating functional and structural connectivity data with neuroreceptor expression patterns, our proposed model of multi-level gradient architecture of the pulvinar complex holds promise for elucidating alterations in pulvinar structure and connectivity observed in various pathological conditions. For instance, reduced availability of D2/3 receptors in the pulvinar correlated with functional connectivity to the superior temporal sulcus and medial occipital lobe and autistic social communication symptoms in individuals with autistic spectrum disorders (*Murayama et al., 2022*). Furthermore, the right anterior pulvinar (PuA), situated at the extreme of our principal coexpression gradient characterized by high expression of 5HTT and low expression of D2 receptors, exhibited reduced volume in patients with Parkinson's disease and comorbid depression. Notably, treatment with serotonergic antidepressants was associated with increased volume of this region, suggesting a protective effect of serotonergic transmission against dopamine depletion-driven degeneration (*Bhome et al., 2022*). In another study involving first-episode psychotic patients, heightened functional connectivity to nodes of the default mode network and central executive network was observed in the dorsal pulvinar. This finding aligns with our multimodal model, wherein increased expression of D2 receptors corresponds to this region (*Kwak et al., 2021*).

We suggest that our integrative, gradient-based model of the organizational features of the pulvinar complex may provide an explorative framework to understand the role of this structure in neuropsychiatric disorders such as Lewy body dementia, Alzheimer's disease, frontotemporal dementia, medial temporal lobe epilepsy, or schizophrenia and other psychotic disorders (*Guye, 2006*; *Anticevic et al., 2015*; *Erskine et al., 2018*; *Capecchi et al., 2020*; *Perez-Rando et al., 2022*; *Velioglu et al., 2023*; *Zhang et al., 2023*), as well as the factors influencing pharmacological treatment response.

Finally, the potential to modulate activity in specific cortical functional networks based on the localization on the main pulvinar axes, as well as its relationships to structural connectivity, offers valuable insights for therapeutic interventions.

This includes deep brain stimulation of the pulvinar, an emerging area of interest in functional neurosurgery for epilepsy, which could benefit from a deeper understanding of pulvinar connectivity and its implications for treatment outcomes (*Kalamatianos et al., 2023*; *Vakilna et al., 2023*; *Wong et al., 2023*). Specifically, the medial (as opposed to the lateral) pulvinar is the desired target due to its connectivity to the medial temporal lobe. Overall, our model opens avenues for both understanding the pathophysiology of neurological and psychiatric disorders and developing targeted therapeutic interventions.

## Future perspectives and limitations

While our study offers valuable insights into the organizational principles of the pulvinar complex, it is not without limitations. One notable limitation of this study lies in the relatively small size of the pulvinar complex compared to other larger cortical or subcortical structures. The high cellular density of the pulvinar poses a challenge for the relatively coarse resolution of currently available imaging techniques. Although the generally high quality of both the main and validation datasets, including rs-fMRI data (*Uğurbil et al., 2013*; *Babayan et al., 2019*), aligns with current standards for imaging investigations of pulvinar connectivity, higher-resolution imaging approaches may offer more granular insights. Advanced techniques, such as ultrahigh-field fMRI, hold promise for uncovering the fine-scale topographical organization of the pulvinar complex. The limitation of small region size is also particularly relevant for the normative PET data that have been resampled to the voxel size of 2 $mm^2$ to match the resolution of BOLD data and suffer from all the inherent limitations of PET imaging in small brain volumes, such as partial volume effects or tracer spillout effects (*Kanel et al., 2023*). In addition, the neurotransmitter normative atlas is derived from group-level averages of data acquired across different centers. As a consequence, the receptor density profiles employed to quantify receptor 'coexpression' are measured across spatially aligned and group-averaged PET scans of different individuals (*Hansen et al., 2022b*), since the radioactive nature of PET and SPECT tracers prevents the study of multiple neurotransmitter systems in the same individual for technical and clinical reasons. On the other hand, this neurotransmitter atlas has been extensively validated both by correlation to an independent autoradiography dataset and to ex vivo gene expression levels measured with microarray techniques (*Hansen et al., 2022a*). As such, it therefore represents, to the best of our knowledge, the most advanced tool available to investigate the relations between neurotransmitter expression patterns in the human brain in vivo. Indeed, as mentioned above, 'gold-standard' reference ex vivo studies on molecular expression levels of the receptors and transporters analyzed in our study, relative to the pulvinar complex, are substantially sparse and nearly absent for the human brain. Therefore, further investigations, possibly by employing immunohistochemical or immunofluorescence microscopy in the human brain coupled with highly sampled gene expression probes, are warranted to validate and enhance our understanding of molecular expression in the human pulvinar.

## Conclusion

This study offers advanced insights into the interplay between pulvinar-cortical and cortico-cortical connectivity in the human brain, along with its structural and molecular correlates. Our proposed model confirms the hypothesis of a 'replication principle', illustrating that the pulvinar complex harbors multiple representations of cortico-cortical functional connectivity organized hierarchically by their information processing significance across its ventro-dorsal and medio-lateral axes. These shared representations traverse pulvinar nuclei continuously, reconciling discrete, individual nuclear connectivity, and coactivation patterns with the idea of a gradient-like organization of pulvinar connectivity.

By systematically assessing the relationship between functional, structural, and molecular aspects of pulvinar organization, we delineate the topographically organized features of the human pulvinar and reveal substantial convergence among the main directives of functional connectivity, anatomical connectivity, and receptor coexpression. Our study underscores the pivotal role of the pulvinar complex in mediating information integration and communication across brain networks at multiple levels of brain organization. We propose that this comprehensive understanding may offer a unified

explanatory framework to enhance comprehension of its involvement in higher-order perceptual and cognitive functions both in healthy patients and in various neuropsychiatric diseases.

# Materials and methods
## Data acquisition and preprocessing
### Primary dataset (HCP)

Structural, diffusion-weighted, and rs-fMRI data of 210 healthy subjects (males = 92, females = 118, age range 22–36 years) were retrieved from the HCP repository (https://humanconnectome.org/). The study protocol was approved by the Washington University in St. Louis Institutional Review Board (IRB) (*Van Essen et al., 2012*).

178 subjects out of 210 (85%) are genetically unrelated. Of the remaining, genetically related subjects, 22 (~10% of the total sample) were included with another subject from the same family group (11 pairs); 6 (3%) were included with two other family members (2 triplets) and 4 (2%) were all parts of the same family group.

MRI data were acquired on a 3T custom-made Siemens 'Connectome Skyra' scanner (Siemens, Erlangen, Germany), equipped with a Siemens SC72 gradient coil.

High-resolution T1-weighted scans (isotropic voxel size = 0.7 mm) were acquired with the following parameters: MP-RAGE sequence, TR = 2400 ms, TE = 2.14 ms (*Van Essen et al., 2012*).

Skull-stripped T1-weighted images were segmented into cortical and subcortical gray matter (GM), white matter (WM), and cerebrospinal fluid (CSF) using FAST and FIRST FSL's tools (*Smith et al., 2004*; *Patenaude et al., 2011*). The MNI space transformations available as part of the minimally preprocessed data were employed (FLIRT 12 degrees of freedom affine; FNIRT nonlinear registration) (*Glasser et al., 2013*).

rs-fMRI data (isotropic voxel size = 2 mm) were acquired with a gradient-echo echo planar imaging (EPI) sequence, using the following parameters: TR = 720 ms, TE = 33.1 ms, 1200 frames, ~15 min/run. Data were acquired separately on different days along two different sessions, each session consisting of a left-to-right (LR) and a right-to-left (RL) phase-encoding acquisition (*Van Essen et al., 2012*; *Uğurbil et al., 2013*; *Smith et al., 2013*). The LR and RL acquisitions of the first session only have been employed in the present work.

Data were retrieved in a minimally preprocessed form which includes artifact and motion correction, registration to 2 mm resolution MNI 152 standard space, high-pass temporal filtering (>2000 s full width at half maximum), ICA-based denoising with ICA-FIX (*Salimi-Khorshidi et al., 2014*), and regression of artifacts and motion-related parameters (*Smith et al., 2013*). Additionally, the global WM and CSF signal was regressed out to further improve ICA-based denoising results (*Plachti et al., 2019*).

Multi-shell DWI data (isotropic voxel size = 1.25 mm) were acquired using a single-shot 2D spin-echo multiband EPI sequence (*Sotiropoulos et al., 2013*). The following acquisition parameters were employed: b-values: 1000, 2000, 3000 mm/s$^2$; 90 directions per shell; spatial isotropic resolution 1.25 mm. DWI scans were retrieved in a minimally preprocessed form including eddy currents, EPI distortion and motion correction, and cross-modal linear registration of structural and DWI images (*Glasser et al., 2013*).

A multi-shell multi-tissue CSD signal modeling was performed to estimate individual response functions in WM, GM, and CSF (*Jeurissen et al., 2014*). Whole-brain, probabilistic tractography (10 million streamlines, iFOD2 algorithm; anatomically constrained tractography) was generated with MRtrix3 default parameters (*Tournier et al., 2010*; *Smith et al., 2012*).

### Validation dataset (LEMON)

Structural, diffusion, and rs-fMRI data of 213 healthy subjects (males = 138, females = 75, age range 20–70 years) were retrieved from the Leipzig Study for Mind-Body-Emotion Interactions (LEMON) dataset (http://fcon_1000.projects.nitrc.org/indi/retro/MPI_LEMON.html). The study protocol was approved by the Ethics Committee of the medical faculty of the University of Leipzig.

MRI data were acquired on a 3T scanner (MAGNETOM Verio, Siemens Healthcare GmbH, Erlangen, Germany) equipped with a 32-channel head coil was employed for MRI data acquisition.

High-resolution T1-weighted (isotropic voxel size = 1 mm) scans were acquired with the following parameters: MP-RAGE sequence, TR = 5000 ms, TE = 2.92 ms. Skull-stripped T1-weighted images were segmented into cortical and subcortical GM, WM, and CSF using FAST and FIRST FSL's tools (*Smith et al., 2004*; *Patenaude et al., 2011*). T1-weighted volumes were also nonlinearly registered to the 1 mm resolution MNI 152 asymmetric template using a FLIRT 12 degrees of freedom affine transform and FNIRT nonlinear registration (*Jenkinson and Smith, 2001*; *Jenkinson et al., 2002*; *Andersson et al., 2007*).

Single-shell DWI data (isotropic voxel size = 1.7 mm) were acquired using a multiband accelerated sequence with a b-value of 1000, and 60 unique diffusion-encoding directions. Data were preprocessed according to the dedicated pipeline implemented via the MRtrix3 software (*Tournier et al., 2019*), which features (1) denoising using Marchenko-Pastur principal component analysis (*Veraart et al., 2016*), (2) removal of Gibbs ringing artifacts (*Kellner et al., 2016*), (3) eddy currents, distortion, and motion correction (*Andersson et al., 2003*; *Smith et al., 2004*; *Andersson and Sotiropoulos, 2016*) and bias field correction using the N4 algorithm (*Tustison et al., 2010*). A single-shell 3-tissue CSD signal modeling was performed using MRtrix3Tissue (*Dhollander et al., 2016*), a fork of MRtrix3 software. Whole-brain, probabilistic tractography (5 million streamlines, iFOD2 algorithm; anatomically constrained tractography) was generated with default parameters (*Tournier et al., 2010*; *Smith et al., 2012*).

For rs-fMRI data (isotropic voxel size = 2.3 mm), a gradient-echo EPI was acquired with the following parameters: TR = 1400 ms, TE = 30 ms, 15.30 min/run (*Babayan et al., 2019*). Data were obtained in a minimally preprocessed form, consisting of the following steps: (1) removal of the first 5 volumes to allow for signal equilibration; (2) motion and distortion correction; (3) outlier and artifact detection (rapidart) and denoising using component-based noise correction (aCompCor); (4) mean-centering and variance normalization of the time series; (5) spatial normalization to 2 mm resolution MNI 152 standard space (*Babayan et al., 2019*; *Mendes et al., 2019*).

## PET data

Volumetric 3D PET images were retrieved from the Network Neurosciences Lab (NetNeuroLab) GitHub page (https://github.com/netneurolab/hansen_receptors/tree/main/data/PET_nifti_images, *Hansen, 2022*). Images were collected for 19 distinct neurotransmitter receptors and transporters in a multicentric data acquisition approach resulting in multiple studies across the world. Images were acquired using optimized imaging preprocessing strategies for each radioligand (*Nørgaard et al., 2019*). In all studies, only healthy participants were scanned, for a total of 1238 healthy individuals (males = 718, females = 520). Full methodologic details about each study, the associated receptor/transporter, tracers, PET cameras, and modeling methods can be found in *Hansen et al., 2022b*. In general, the measured estimates for each image (binding potential and/or tracer distribution volume) are proportional to receptor density, and then we will refer to them as 'receptor density maps' for simplicity.

Receptor density maps, originally available at different resolutions, were resliced to the 2 mm MNI152 standard template resolution and converted to z-scores. Four receptor density maps (5HT1b, D2, mGluR5, vAChT) were acquired using the same tracer in different studies and were converted to a weighted average as in the reference paper. Ten receptor/transmitters (5HT1a, 5HT1b, 5HTT, CB1, D2, DAT, GABA-A, MOR, and NET) were acquired with different tracers in different studies; these receptor maps were not averaged but kept as separated, resulting in a total of 29 receptor density maps. *Supplementary file 1* summarizes details about the neurotransmitter, tracers, and number of subjects for each study.

## Diffusion map embedding

We derived functional connectivity, structural connectivity, and receptor expression gradients for the left and right pulvinar separately. All the analyses were performed in standard space (MNI152 2006 template, 2 mm voxel size). Voxels belonging to the left or right pulvinar were defined according to the Automated Anatomical Labeling v3 Atlas (AAL3, 2 mm version) (*Rolls et al., 2020*), while cortical ROIs were derived from the 400-areas version of the local-global cortical parcellation proposed by *Schaefer et al., 2018*.

Individual functional connectomes were obtained from Pearson's correlation of BOLD signal in each pulvinar voxel to the average signal within each cortical ROI. As in similar work, negative correlation values were zeroed and sparsity sampling was applied by row-wise thresholding of top 10% connectivity values, with all the values below the 90th percentile set to 0 (*Margulies et al., 2016*; *Katsumi et al., 2023*). Functional connectivity matrices were then normalized with Fisher's r-to-z transformation and averaged across all subjects (and across LR and RL sessions, for the HCP dataset) to obtain a group-level dense functional connectome.

Structural connectomes were obtained after registration of each whole-brain tractogram to the MNI space. Streamlines connecting the pulvinar to each cortical ROI were extracted from the whole-brain tractogram and mapped back to their pulvinar endpoint voxels using the *tckmap* command in MRtrix3, to obtain individual, voxel-wise pulvinar tract-density distribution maps. Such individual maps were concatenated for all cortical ROIs, converted to voxel-to-cortical ROI tract-density matrices, and averaged across all subjects to obtain a group-level dense structural connectome.

Finally, the normalized receptor density was sampled for each voxel within the pulvinar by concatenating. This process resulted in a final pulvinar voxel-by-receptor density matrix, representing the expression levels of 19 neurotransmitters or receptors for each voxel within the pulvinar. This matrix will be referred to as the 'coexpression matrix'.

We applied diffusion embedding (*Coifman and Lafon, 2006*), an unsupervised learning algorithm widely employed in the field of gradient mapping (*Guell et al., 2018*; *Tian et al., 2020*; *Hong et al., 2020*; *Katsumi et al., 2023*). Following previous work, each group's asymmetric feature matrix (functional dense connectome, structural dense connectome, coexpression matrix) was converted to a symmetric voxel-by-voxel cosine distance similarity matrix (1-minus-cosine distance). For diffusion embedding, an alpha value of 0.5 was employed to maximize robustness to noise. The source code for this method is available at https://github.com/sensein/mapalign (*Ghosh, 2022*). Eigenvalues were utilized to quantify the variance explained by each gradient, and scree plots of explained variance against the number of gradients were employed to select the most appropriate number of gradients for each side (left or right) and modality. To address the intrinsic sign indeterminacy of gradient representations, right-sided gradients were sign-flipped to align with their left-sided counterparts if needed.

Functional and structural connectivity gradients were also obtained from cortico-cortical connectivity matrices and employed for further analysis. Functional connectivity was computed as the pairwise Pearson's correlation between the average time series of the 400 cortical ROIs. For structural connectivity gradient embedding, the raw number of streamlines connecting each pair of cortical ROIs was extracted from the whole-brain tractogram separately for the left and right hemispheres, to emphasize intra-hemispheric patterns of connectivity. Right hemisphere gradients were sign-flipped to align with their left hemisphere counterparts if necessary. The diffusion embedding analysis was performed using the same methods and parameters as for the pulvinar gradients. A graphical overview of the gradient mapping protocol is presented in *Figure 1*.

## Gradient analysis and statistics

To facilitate the interpretation of their significance, we performed post hoc characterization of each pulvinar gradient at various levels.

First, we assessed the involvement of discrete pulvinar nuclei in gradient organization by calculating the distribution of gradient values for each nucleus. Histological nuclei were identified using the AAL atlas, which incorporates digitalized parcellation of thalamic nuclei derived from postmortem histology (*Iglesias et al., 2018*). Furthermore, to explore whether the continuous gradient architecture derived from diffusion embedding could be mapped to discrete units overlapping with pulvinar nuclei, we employed a k-means clustering strategy on gradient values, as described in previous work (*Guell et al., 2020*). The relevant gradients to be included in the clustering analysis were determined by identifying the elbow in their explained variance percentage graph (as in the paragraph below). For both the left and right pulvinar, the relevant gradients were then normalized within a range from 0 to 1 and concatenated. Various iterations of k-means clustering were then applied in the resulting gradient space, ranging from k=2 to k=30, and the silhouette coefficient (*Rousseeuw, 1987*) was utilized to identify the optimal number of clusters.

Gradient clustering was performed separately for left and right pulvinar, both after concatenating relevant gradients from each single imaging modality alone (functional connectivity, structural connectivity, receptor coexpression) or by concatenating together the most relevant gradients from all the imaging modalities (combined clustering). For each single-modality and combined cluster, the similarity to anatomical pulvinar nuclei was quantified using the Dice similarity coefficient (*Dice, 1945*).

To analyze the functional or structural connectivity gradients, we obtained gradient-weighted cortical connectivity maps by multiplying each voxel's connectivity to the 400 cortical ROIs by the corresponding gradient value. These voxel-wise, gradient-weighted pulvinar-cortical connectivity maps were then averaged to derive a single cortical projection for each pulvinar connectivity gradient.

For what concerns gradient-weighted functional connectivity maps that are symmetric and highly similar between left and right hemisphere, left and right pulvinar-cortical connectivity maps were averaged for visualization purposes. Regarding structural connectivity maps, due to the asymmetrical distribution of pulvinar structural connectivity, that is biased toward the ipsilateral cortical hemisphere, we flipped the right gradient-weighted connectivity maps on the x axis before averaging left and right pulvinar-cortical connectivity maps, so that all the ipsilateral cortical connectivity is shown on the left side and all the contralateral connectivity is shown on the right side.

To further understand the involvement of cortical networks in each of the cortical projections of structural and functional cortical gradients, we examined the distribution of gradient-weighted connectivity values across the seven major functional cortical networks, as defined in *Thomas Yeo et al., 2011*.

Additionally, we explored the relationship between pulvinar-cortical and cortico-cortical connectivity gradients by calculating Pearson's correlation for each gradient-weighted functional connectivity map with each cortico-cortical gradient map.

Receptor coexpression gradients were characterized by correlating voxel-wise gradient values to normalized receptor density distributions.

Finally, to investigate the relation between pulvinar gradients obtained from different modalities, we computed the pairwise Pearson's correlation between gradients obtained with different modalities.

To account for the SA properties of gradient maps, for all the correlations described, statistical significance was assessed using the permutational approach described in *Burt et al., 2020*. Briefly, this method takes as input geometric distance matrices for SA estimation and involves the generation of a given number of SA-preserving permuted surrogate maps, which are then employed as nulls to estimate a permutational null distribution of the test statistic (*Burt et al., 2020*). Pairwise Euclidean distances between left or right pulvinar voxel coordinates were employed for pulvinar null models, while for cortical parcellated connectivity data Euclidean distances were estimated between centroids of each cortical ROI. In both cases, 1000 surrogates were generated to estimate the null distribution. Statistical tests were controlled for FDR using Benjamini and Hochberg's correction.

## Reliability and reproducibility assessment

We conducted several analyses to assess the robustness and consistency of our main findings. Specifically, both functional and structural connectivity gradients underwent split-half and test-retest replicability analysis, as well as reproducibility analysis.

For split-half replicability analysis, the main dataset was segmented into two halves, with 105 subjects each. Subjects were randomly assigned to each half, and 100 instances of the shuffling algorithm were performed. In each iteration, individual connectivity matrices of each half were averaged into dense connectomes and underwent diffusion embedding. The average Pearson's correlation for each gradient across all the iterations was used as a reliability measure.

Test-retest replicability was based on an extension of the HCP dataset, consisting of 44 subjects (males = 13; females = 31; age range: 22–36 years) with available test-retest structural and diffusion acquisition. The dense structural and functional connectomes obtained from the test and retest datasets were compared using Pearson's correlation as a measure of similarity.

Finally, reproducibility was determined by recomputing results in the validation dataset and correlating pulvinar gradients between the primary and replication datasets.

In all the mentioned cases, gradients were realigned before correlation by using Procrustes' analysis to obtain more reliable estimates of gradient similarity (*Hong et al., 2020*). This approach ensured

that the gradients from different datasets were aligned properly before assessing their similarity, enhancing the robustness of our analyses.

To evaluate the possible influence of SNR on connectivity-derived diffusion embeddings, we have performed a voxel-wise, modality-specific, SNR assessment to investigate correlation between spatial distribution of noise and diffusion embeddings. For each subject, we separately calculated voxel-wise SNR maps for the left and right pulvinar, using both functional (BOLD) volumes and DWI data. For BOLD volumes, we employed the widely accepted definition of temporal signal-to-noise ratio (tSNR) (**Murphy et al., 2007**)

$$tSNR = \frac{t_{mean}}{t_{std}}$$

where $t_{mean}$ and $t_{std}$ are, respectively, the mean and the standard deviation of each voxel's signal across the time series.

For DWI data, we employed a similar definition (**Cai et al., 2021**) that allows estimation of SNR from multiple b=0 diffusion-weighted volumes:

$$SNR = \frac{S_{mean}}{S_{std}}$$

where S is the voxel's signal intensity, and the mean ($S_{mean}$) and standard deviation ($S_{std}$) were computed across all the b0-weighted volumes (18 for HCP dataset; 7 for LEMON dataset). Individual pulvinar SNR maps were then averaged to generate group-level estimates of SNR spatial distribution. The resulting, modality-specific average SNR maps were correlated with the diffusion gradients derived from the corresponding modality, following the same approach described in the previous section (Pearson's correlation; p-values corrected using spatial null models for SA, and Benjamini-Hochberg correction for FWE).

# Additional information

## Funding

| Funder | Grant reference number | Author |
| --- | --- | --- |
| Ministero della Salute | Current Research Funds | Augusto Ielo<br>Lilla Bonanno<br>Angelo Quartarone |
| Ministero dell'università e della ricerca | Project Code 0000006 | Alberto Cacciola |

The funders had no role in study design, data collection and interpretation, or the decision to submit the work for publication.

## Author contributions

Gianpaolo Antonio Basile, Conceptualization, Data curation, Formal analysis, Investigation, Visualization, Methodology, Writing – original draft; Augusto Ielo, Data curation, Software, Formal analysis, Investigation, Visualization, Methodology, Writing – review and editing; Lilla Bonanno, Data curation, Methodology, Writing – review and editing; Antonio Cerasa, Data curation, Methodology, Writing – original draft, Writing – review and editing; Giuseppe Santoro, Resources, Data curation, Investigation, Writing – review and editing; Demetrio Milardi, Resources, Data curation, Formal analysis, Visualization, Writing – review and editing; Giuseppe Pio Anastasi, Conceptualization, Supervision, Visualization, Writing – review and editing; Ambra Torre, Data curation, Formal analysis, Investigation, Methodology; Sergio Baldari, Riccardo Laudicella, Investigation, Methodology, Writing – review and editing; Michele Gaeta, Maria Pina Serra, Marcello Trucas, Investigation, Visualization, Methodology, Writing – review and editing; Marina Quartu, Conceptualization, Investigation, Visualization, Methodology, Writing – review and editing; Angelo Quartarone, Funding acquisition, Investigation, Writing – review and editing; Manojkumar Saranathan, Conceptualization, Data curation, Visualization, Methodology, Writing – review and editing; Alberto Cacciola, Conceptualization, Data curation, Formal

analysis, Supervision, Funding acquisition, Investigation, Visualization, Methodology, Writing – original draft, Project administration, Writing – review and editing

### Author ORCIDs
Marina Quartu  https://orcid.org/0000-0002-1884-3597
Alberto Cacciola  https://orcid.org/0000-0001-9412-4116

### Ethics
Human subjects: The primary dataset (HCP) was provided by the Human Connectome Project, WU-Minn Consortium (Principal Investigators: David Van Essen and Kamil Ugurbil; 1U54MH091657). The data are openly available from https://humanconnectome.org. The study protocol was approved by the Washington University in St. Louis Institutional Review Board (IRB). The "Leipzig Study for Mind-Body-Emotion Interactions" (LEMON) data used as a validation dataset was provided by the Mind-Body-Emotion group at the Max Planck Institute for Human Cognitive and Brain Sciences. The data are openly available from http://fcon_1000.projects.nitrc.org/indi/retro/MPI_LEMON.html. The study protocol was approved by the Ethics Committee of the medical faculty of the University of Leipzig.

Reviewer #1 (Public review): https://doi.org/10.7554/eLife.100937.3.sa1
Reviewer #2 (Public review): https://doi.org/10.7554/eLife.100937.3.sa2
Author response https://doi.org/10.7554/eLife.100937.3.sa3

---

## Additional files

### Supplementary files
Supplementary file 1. Information on the neurotransmitter, tracers, and number of subjects for each study.

MDAR checklist

### Data availability
The primary dataset (HCP) was provided by the Human Connectome Project, WU-Minn Consortium (Principal Investigators: David Van Essen and Kamil Ugurbil; 1U54MH091657). The data are openly available from https://humanconnectome.org. The "Leipzig Study for Mind-Body-Emotion Interactions" (LEMON) data used as a validation dataset was provided by the Mind-Body-Emotion group at the Max Planck Institute for Human Cognitive and Brain Sciences. The data are openly available from http://fcon_1000.projects.nitrc.org/indi/retro/MPI_LEMON.html. The code and maps obtained in the present work are available at GitHub (copy archived at *BrainMappingLab, 2025*).

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
