## [Editor Report · eLife Assessment]

This study presents a **useful** characterisation of the topographical organisation of the human pulvinar, an associative thalamic subregion crucial for visual perception and attention. The evidence supporting the conclusions is **solid** given the multimodal validation and replication across datasets, although even higher-resolution imaging data would have strengthened the study. In their revised manuscript, the authors elaborated further on the motivation for their study and conducted several robustness checks. Nevertheless, there remains an opportunity for a more fully integrated interpretation of the findings. The work would be of interest to neuroscientists, neurologists, and neuropsychiatrists working on pulvinar functioning in health and disease.

---

## [Referee Report · Reviewer #1 (Public review)]

Summary:

The current work explored the link between the pulvinar intrinsic organisation and its functional and structural connectivity patterns of the cortex using different dimensional reduction techniques. Overall they find relationships between pulvinar-cortical organization and cortico-cortical organization, and little evidence for clustered organization. Moreover they investigate PET maps to understand how neurotransmitter/receptor distributions vary within the pulvinar and along its structural and functional connectivity axes.

Strengths:

(1) There is a replication dataset and different modalities are compared against each other to understand the structural and functional organisation of the pulvinar complex

In their revision, the authors further detailed the motivation of their study and performed various robustness checks, answering my concerns. Nevertheless, further work is needed to fully understand the role of the pulvinar nuclei and the rest of the thalamic nuclei as well as the rest of the brain, including more diverse datasets and techniques.

---

## [Referee Report · Reviewer #2 (Public review)]

Summary:

The authors aimed to explore and better understand the complex topographical organization of the human pulvinar, a brain region crucial for various high-order functions such as perception and attention. They sought to move beyond traditional histological subdivisions by investigating continuous 'gradients' of cortical connections along the dorsoventral and mediolateral axes. Using advanced imaging techniques and a comprehensive PET atlas of neurotransmitter receptors, the study aimed to identify and characterize these gradients in terms of structural connections, functional coactivation, and molecular binding patterns. Ultimately, the authors targeted to provide a more nuanced understanding of pulvinar anatomy and its implications for brain function in both healthy and diseased states.

Strengths:

A key strength of this study lies in the authors' effort to comprehensively combine multimodal data, encompassing both functional and structural connectomics, alongside the analysis of major neurotransmitter distributions. This approach enabled a more nuanced understanding of the overarching organizational principles of the pulvinar nucleus within the broader context of whole-brain connectivity. By employing cortex-wide correlation analyses of multimodal embedding patterns derived from 'gradients,' which provide spatial maps reflecting the underlying connectomic and molecular similarities across voxels, the study offers a thorough characterization of the functional neuroanatomy of the pulvinar.

Weaknesses:

Despite its strengths, the current manuscript falls short in presenting the authors' unique perspectives on integrating the diverse biological principles derived from the various neuroimaging modalities. The findings are predominantly reported as correlations between different gradient maps, without providing the in-depth interpretations that would allow for a more comprehensive understanding of the pulvinar's role as a central hub in the brain's network.

---

## [Author Response]

The following is the authors’ response to the original reviews

**Public Reviews:**

**Reviewer #1 (Public review):**
Summary:The current work explored the link between the pulvinar intrinsic organisation and its functional and structural connectivity patterns of the cortex using different dimensional reduction techniques. Overall they find relationships between pulvinar-cortical organization and cortico-cortical organization, and little evidence for clustered organization. Moreover, they investigate PET maps to understand how neurotransmitter/receptor distributions vary within the pulvinar and along its structural and functional connectivity axes.Strengths:There is a replication dataset and different modalities are compared against each other to understand the structural and functional organisation of the pulvinar complex.Weaknesses:(1) What is the motivation of the study and how does this work extend previous assessments of the organization of the complete thalamus within the gradient framework?

Thank you for raising this central question. As already mentioned in the main text, pulvinar is one of the largest and prototypical associative nuclei, yet its organizational principles in the human brain remain relatively unexplored. The substantial body of anatomical research conducted in primate species suggests the coexistence of multiple coexisting and overlapping corticotopic representations on the pulvinar complex.

Existing connectivity-based parcellation studies of pulvinar organization often overlook these organizational principles, as the resulting parcellation may reflect a linear combination of single overlapping connectopies rather than accurately capturing their distinct and unique spatial arrangement.

Investigations of thalamic connectivity have already revealed overarching organizational principles within the thalamus, which are partially reflected in its cytoarchitecture subdivision. These principles are associated with core and matrix thalamic neuronal subpopulation, and their distinct contributions to large-scale connectivity networks.

Since gradient selection relies on the explained variance of the diffusion embeddings, and pulvinar-cortical connectivity likely accounts for only a limited portion of the variance in thalamocortical connectivity, we chose to focus specifically on the pulvinar nucleus. This approach was intended to ensure that the local connectivity principles of the pulvinar are not overshadowed by the broader connectotopical organization of the entire thalamus.

This rationale aligns with findings in topographically organized regions of the cerebral cortex, such as M1, S1 or visual areas. In these regions, distinct principles of topographical organization are not readily apparent when analyzing whole-brain connectivity embedding but emerge when dimensionality reduction is applied to region-specific connectivity data.

(2) Why is the current atlas chosen for the delineation of the pulvinar and individualized maps not considered? Given the size of the pulvinar, more validation of the correctness of the atlas may be helpful.

To improve signal-to-noise ratio and in alignment with previous studies, we performed diffusion embedding on the group-level, averaged connectivity matrices rather than estimating gradients at the individual subject level.

The decision to use a standard-space atlas for pulvinar delineation, rather than individualized parcellation, was driven by technical considerations: (1) functional MRI data were already transformed to MNI space; and (2) individualized parcellation of thalamic nuclei can result in varying pulvinar volumes across subjects, complicating the averaging of connectivity data. By using a standard-space atlas, we ensured that connectivity was consistently extracted from the same set of voxels across all subjects.

We selected the AAL3 atlas (Rolls et al., 2020)over other existing thalamic atlases for practical reasons: the atlas incorporates an ex-vivo thalamic parcellation (Iglesias et al., 2018) with a specific delineation of pulvinar nuclei, which was necessary for subsequent analyses. In the revised version of the manuscript, to validate our findings, we replicated the pulvinar gradient using a different pulvinar delineation from a recent, thalamus-specific atlas (Su et al., 2019). Notably, the spatial distribution of pulvinar connectivity and coexpression gradients remained consistent, regardless of the choice of the thalamic atlas, underscoring the robustness of our results.

(3) Overall the study feels a little incremental and a repetition of what others have done already in the thalamus. It would be good to know how focusing only on the pulvinar changes interpretation, for example by comparing thalamic and pulvinar gradients?

The authors acknowledge the existing body of literature that has examined thalamic connectivity under the lens of the connectivity gradient framework. While these studies may provide valuable insights into the functional topography of the pulvinar complex -given its prominent role within the thalamus - we contend that a focused analysis of pulvinar connectivity offers a unique opportunity to uncover the specific organization principles of this nuclear complex. By isolating the pulvinar, we aimed to avoid the potential overshadowing of its local connectivity patterns by the broader connectotopical organization of the entire thalamus. However, as we believe that our findings are best interpreted within the broader context of general thalamic connectivity organization, we have included an additional paragraph in the Discussion section, which explores the similarities and differences between thalamic and pulvinar gradients, offering a more integrative perspective on our results.

“In recent years, different works have explored the spatial arrangement of thalamic connectivity within a connectivity gradient framework. Diffusion embedding of thalamocortical functional connectivity has revealed a principal, medio-lateral gradient that was found correlated to thalamic structural subdivisions, and a secondary, antero-posterior gradient associated with thalamic functional subfields, and showing progression from unimodal sensorimotor cortical networks to multimodal attention and associative networks. Interestingly, the principal thalamic gradient shows a medio-lateral arrangement on the pulvinar axis while the secondary gradients correspond more to a ventral-dorsal pulvinar axis (Yang et al. 2020). In particular, further independent investigations have suggested that the progressing pattern of thalamic connectivity from unimodal to transmodal cortices is strongly associated to the local density of core and matrix cell types, thus establishing a link between molecular properties and functional connectivity dynamics (Müller et al. 2020; Huang et al. 2024). Our findings complement and expand the existing literature by revealing a similar arrangement of cortical connectivity patterns on the pulvinar complex, and elucidating its relationship to in-vivo estimates of molecular markers of neurotransmission. We found that the gradient associated to unimodal-transmodal cortical connectivity accounted for the highest percentage of variance of variance in cortico-pulvinar connectivity, in line with its well-acknowledged role of associative nucleus. It is noteworthy that, in analyses of thalamocortical gradients, the pulvinar complex is situated towards the “sensorimotor” extreme of the unimodal-to-transmodal thalamic gradient (Yang et al., 2020). This likely reflects its prominent connectivity to visual and sensory areas compared to other thalamic nuclei. Nevertheless, the extensive and intricate association of pulvinar with multiple cortical networks emerges is strongly evident in various functional connectivity investigations (Basile et al., 2021; Kumar et al., 2017, 2022). By isolating pulvinar-cortical from broader thalamocortical connectivity, our analysis was able to provide additional insights into the spatial organization of its connectivity with different cortical networks, highlighting the pulvinar's remarkable functional diversity and complexity.”

(4) Could it be that the gradient patterns stem from lacking anatomical and functional resolutions (or low SNR) therefore generating no sharp boundaries?

The gradient organization described in our results is aligns with anatomical evidence on non-human primates (Shipp, 2003), and with existing neuroimaging studies in humans, which report limited correspondence between connectivity-based hard clustering solutions and histological delineation of pulvinar nuclei. However, we recognize the critical importance of assessing the impact of SNR on connectivity measures derived from functional and structural MRI. In the revised manuscript, we have included an additional analysis to investigate the potential impact of local noise on gradient reconstruction. This analysis involved sampling voxel-wise SNR estimates in the pulvinar from both BOLD and diffusion-weighted MRI data, averaging these estimates to generate group-level, modality-specific SNR maps. We then assessed spatial correlations between these maps and the gradient embeddings using the same methodological framework employed throughout the study. Our findings indicate that functional connectivity gradients are weakly, but significantly correlated to SNR, with the strongest correlation observed for the third gradient (left hemisphere G_FC_1 r = -0.30, SA-corrected p < 0.001, G_FC_2 r = 0.22, SA-corrected p = 0.05, G_FC_3 r = 0.55, SA-corrected p < 0.001; right hemisphere G_FC_1 r = -0.41, SA-corrected p < 0.001, G_FC_2 r = 0.22, SA-corrected p = 0.008, G_FC_3 r = 0.52, SA-corrected p = 0.017). In contrast, structural connectivity gradients showed no significant correlation with SNR (left hemisphere G_SC_1 r = 0.06, SA-corrected p = 0.82, G_SC_2 r = -0.33, SA-corrected p = 0.01; right hemisphere G_SC_1 r = 0.40, SA-corrected p = 0.28, G_SC_2 r = -0.19, SA-corrected p = 0.31).

**Reviewer #1 (Recommendations for the authors):**
(1) Please add more literature on thalamus gradients and interpret this with care.

Thank you for the suggestion. We have added the following paragraph in the Discussion section:

“In recent years, different works have explored the spatial arrangement of thalamic connectivity within a connectivity gradient framework. Diffusion embedding of thalamocortical functional connectivity has revealed a principal, medio-lateral gradient that was found correlated to thalamic structural subdivisions, and a secondary, antero-posterior gradient associated with thalamic functional subfields, and showing progression from unimodal sensorimotor cortical networks to multimodal attention and associative networks. Interestingly, the principal thalamic gradient shows a medio-lateral arrangement on the pulvinar axis while the secondary gradients correspond more to a ventral-dorsal pulvinar axis (Yang et al. 2020). In particular, further independent investigations have suggested that the progressing pattern of thalamic connectivity from unimodal to transmodal cortices is strongly associated to the local density of core and matrix cell types, thus establishing a link between molecular properties and functional connectivity dynamics (Müller et al. 2020; Huang et al. 2024). Our findings complement and expand the existing literature by revealing a similar arrangement of cortical connectivity patterns on the pulvinar complex, and elucidating its relationship to in-vivo estimates of molecular markers of neurotransmission. We found that the gradient associated to unimodal-transmodal cortical connectivity accounted for the highest percentage of variance of variance in cortico-pulvinar connectivity, in line with its well-acknowledged role of associative nucleus. It is noteworthy that, in analyses of thalamocortical gradients, the pulvinar complex is situated towards the “sensorimotor” extreme of the unimodal-to-transmodal thalamic gradient (Yang et al., 2020). This likely reflects its prominent connectivity to visual and sensory areas compared to other thalamic nuclei. Nevertheless, the extensive and intricate association of pulvinar with multiple cortical networks emerges is strongly evident in various functional connectivity investigations (Basile et al., 2021; Kumar et al., 2017, 2022). By isolating pulvinar-cortical from broader thalamocortical connectivity, our analysis was able to provide additional insights into the spatial organization of its connectivity with different cortical networks, highlighting the pulvinar's remarkable functional diversity and complexity.

As regards structural connectivity, existing accounts describe a medio-lateral organization of thalamocortical connections, corresponding to an antero-posterior gradient on the cortical mantle. This gradient organization appears to be anchored to genetic markers of different cell types (Oldham and Ball 2023). In line with their findings, we describe a principal axis of structural connectivity in the pulvinar complex that is arranged on the mediolateral axis, and we enforce the notion of a deep relationship between structural connections and molecular expression of neurotransmission markers. On the other hand, the patterns of connectivity with the cerebral cortex do not correspond to a clear antero-posterior axis on the cerebral cortex, probably showing the predominance of local connectivity over the global thalamic structural topography. Further investigations are warranted to ascertain whether the structural gradients of the pulvinar complex may be in continuity with this general cortico-thalamic connectivity gradient.”

(2) Please state the motivation of the work more clearly and what makes it different from related literature.

Thank you for pointing us to this lack of clarity. We have added the following paragraph in the Introduction section:

“In particular, investigations of thalamic connectivity within the gradient framework have uncovered general organizational principles within the thalamus, which are partially reflected in thalamic cytoarchitecture subdivisions. These principles have been linked to core and matrix thalamic neuronal subpopulation, and to their differential contribution to large-scale connectivity networks (Müller et al., 2020; Yang et al., 2020). However, given the remarkable functional complexity and diversity of the pulvinar complex, these global spatial organization patterns likely capture only part of its functional topography. With this in mind, isolating pulvinar connectivity from the remaining thalamocortical connectome would ensure that local organizational principles are not obscured by the global connectotopic structure of the entire thalamus.”

(3) Why did the authors opt for a whole brain labelling atlas, would a thalamus-specific atlas not be more suitable?

Despite being a large-scale whole brain atlas, the labeling atlas of choice (AAL3) incorporates a thalamus-specific parcellation from previous work (Iglesias et al., 2018), derived from ex-vivo data and including subdivision of the pulvinar complex into anterior, inferior, lateral and medial nuclei. In the revised version of the manuscript, to validate our findings, we replicated the pulvinar gradient using a different pulvinar delineation from a recent, thalamus-specific atlas (Su et al., 2019). We show these results in Supplementary Figure 1. Notably, the spatial distribution of pulvinar connectivity and coexpression gradients remained consistent, regardless of the choice of the thalamic atlas, underscoring the robustness of our results.

(4) How did the authors account for the potential low sensitivity of subcortical signals in the PET data?

We acknowledge the inherent limitations in spatial sensitivity that are a common drawback of PET imaging. However, the PET data employed in the present study were derived from a high-quality dataset collected across multiple studies, predominantly acquired using high resolution scanners (Hansen et al., 2022; see supplementary material at https://static-content.springer.com/esm/art%3A10.1038%2Fs41593-022-01186-3/MediaObjects/41593_2022_1186_MOESM3_ESM.xlsx for technical details). Furthermore, the reliability of neurotransmission markers measurements at the subcortical level has been validated against genetic transcription markers (Hansen, Markello, et al., 2022; Hansen, Shafiei, et al., 2022), ensuring robust and biologically meaningful results.

(5) What about SNR of the metrics within the pulvinar?

The referee raises a crucial and complex point, prompting us to conduct additional analyses. We recognize the critical importance of assessing the impact of SNR on connectivity measures derived from functional and structural MRI. In the revised manuscript, we have included an additional analysis to investigate the potential impact of local noise on gradient reconstruction. Therefore, we have incorporated the following text into the manuscript:

Results (5. Reliability and Reproducibility):

“To assess the influence of local noise on functional and structural connectivity gradients, we calculated the spatial correlation between gradient values and averaged voxel-wise estimates of signal-to-noise ratio (SNR) from functional and structural MRI data, respectively. We found that functional connectivity gradients are weakly, but significantly correlated with the SNR, with the strongest correlation observed for the third gradient (left hemisphere G_FC_1 r = -0.30, SA-corrected p < 0.001, G_FC_2 r = 0.22, SA-corrected p = 0.05, G_FC_3 r = 0.55, SA-corrected p < 0.001; right hemisphere G_FC_1 r = -0.41, SA-corrected p < 0.001, G_FC_2 r = 0.22, SA-corrected p = 0.008, G_FC_3 r = 0.52, SA-corrected p = 0.017). In contrast, structural connectivity gradients were not significantly associated with SNR (left hemisphere G_SC_1 r = 0.06, SA-corrected p = 0.82, G_SC_2 r = -0.33, SA-corrected p = 0.01; right hemisphere G_SC_1 r = 0.40, SA-corrected p = 0.28, G_SC_2 r = -0.19, SA-corrected p = 0.31) (Supplementary Figure 5).”

Methods (4. Reliability and reproducibility assessment):

“To evaluate the possible influence of SNR on connectivity-derived diffusion embeddings, we have performed a voxel-wise,

modality-specific, SNR assessment to investigate correlation between spatial distribution of noise and diffusion embeddings. For each subject, we separately calculated voxel-wise SNR maps for the left and right pulvinar, using both functional (BOLD) volumes and DWI data. For BOLD volumes, we employed the widely accepted definition of temporal signal to noise (tSNR) (Murphy et al., 2006):\begin{document}$$\displaystyle t S N R=\frac{t_{\text {mean }}}{t_{\text {std }}}$$\end{document}

where T_mean_ and T_std_ are, respectively, the mean and the standard deviation of each voxel’s signal across the time series.

For the DWI data, we applied a similar approach (Cai et al., 2021) that allows estimation of SNR from multiple b=0 diffusion weighted volumes:\begin{document}$$\displaystyle S N R=\frac{S_{\text {mean }}}{S_{\text {std }}}$$\end{document}

where S is the voxel’s signal intensity, and the mean (*Smean*) and standard deviation (*Sstd*) were computed across all the b0-weighted volumes (18 for HCP dataset; 7 for LEMON dataset). Individual pulvinar SNR maps were then averaged to generate group-level estimates of SNR spatial distribution. The resulting, modality-specific average SNR maps were correlated with the diffusion gradients derived from the corresponding modality, following the same approach described in the previous section (Pearson’s correlation; p-values corrected using spatial null models for spatial autocorrelation, and Benjamini-Hochberg correction for FWE).”

(6) The numbers of the screeplot / numbers in figures are quite small and not so easy to read.

Thank you for highlighting this point. We have fixed this issue in the revised version of the Figures.

(7) How do you know the pulvinar mask is not also picking up on the cortical spinal tract?

To ensure that pulvinar masks did not pick up streamlines from the corticospinal tracts, we performed a thorough visual inspection of the tractograms that were employed for structural connectivity estimation. For each subject-specific tractogram, we randomly subsampled 10000 streamlines after transformation into MNI standard space and summed up these results to generate a group-level tractogram in standard space. The resulting track-density images (Author response image 1) demonstrate only minimal involvement of descending/ascending tracts from/to the brainstem and spinal cord, confirming the specificity of the pulvinar masks.

**Author response image 1. sa3fig1:** Group-level structural connectivity of the pulvinar complex. Track-density images have been normalized and overlaid on the MNI152 standard template.

(8) There is no mention of the within pulvinar gradients that then are correlated with PET patterns or across gradients are tested to spatial autocorrelation? I believe it is only mentioned for the cortex.

Thanks for providing us with the opportunity to clarify this important aspect, which is mentioned in the Methods section (3. Gradient analysis and statistics):

“To account for the spatial autocorrelation (SA) properties of gradient maps, for all the correlations described, statistical significance was assessed using the permutational approach described in Burt et al. (2020). Briefly, this method takes as input geometric distance matrices for SA estimation and involves the generation of a given number of SA-preserving permuted surrogate maps, which are then employed as nulls to estimate a permutational null distribution of the test statistic (Burt et al. 2020). Pairwise Euclidean distances between left or right pulvinar voxel coordinates were employed for pulvinar null models, while for cortical parcellated connectivity data Euclidean distances were estimated between centroids of each cortical ROI. In both cases, 1000 surrogates were generated to estimate the null distribution. Statistical tests were controlled for false discovery rate (FDR) using Benjamini and Hochberg’s correction.”

However, to enhance readability, we have highlighted this concept in the Results section (3. The unimodal-to-transmodal gradient (G_FC_1) aligns with receptor expression on the dorso-ventral pulvinar axis):

“To take into account the effects of spatial autocorrelation, we corrected the resulting p-values using a method based on SA-preserving spatial null models (Burt et al. 2020)”.

(9) I don't fully understand why the mappings are so patchy of the structural connectivity gradient? Maybe some normalisation went wrong? Other papers on thalamic gradients show smoother patterns.

We thank the Reviewer for the observation. After thoroughly reviewing the related codes, we found no normalization errors. However, we identified a visualization issue, which has been addressed in the revised version. Specifically, the structural gradient representations showed in the figures were based on the averaged values of left and right pulvinar gradients both of which include structural connectivity to either the ipsilateral or contralateral cerebral cortex. Since ipsilateral connectivity is more prominently represented than contralateral connectivity, this led to asymmetric gradient patterns between ipsilateral and contralateral cortical gradients, resulting in a patchy representation when gradients were averaged between left and right pulvinar. To resolve this, we adjusted the visualization by flipping the right pulvinar gradient representations along the x axis, aligning all the ipsilateral cortical connectivity on the left side and all the contralateral connectivity on the right. This adjustment produced smoother, more readable, and interpretable visualizations. Additionally, it allowed the asymmetry between ipsilateral and contralateral connections to be more clearly appreciated.

(10) The final statement of the abstract is misleading as we at this point don't know how making spatial pattern maps in the pulvinar may help understand the role of the pulvinar in health and disease.

We appreciate the Reviewer’s suggestion and have updated the expression accordingly:

“Our findings represent a significant step forward in advancing the understanding of pulvinar anatomy and function, offering an exploratory framework to investigate the role of this structure in both health and disease.”

**Reviewer #2 (Public review):**
Summary:The authors aimed to explore and better understand the complex topographical organization of the human pulvinar, a brain region crucial for various high-order functions such as perception and attention. They sought to move beyond traditional histological subdivisions by investigating continuous 'gradients' of cortical connections along the dorsoventral and mediolateral axes. Using advanced imaging techniques and a comprehensive PET atlas of neurotransmitter receptors, the study aimed to identify and characterize these gradients in terms of structural connections, functional coactivation, and molecular binding patterns. Ultimately, the authors targeted to provide a more nuanced understanding of pulvinar anatomy and its implications for brain function in both healthy and diseased states.Strengths:A key strength of this study lies in the authors' effort to comprehensively combine multimodal data, encompassing both functional and structural connectomics, alongside the analysis of major neurotransmitter distributions. This approach enabled a more nuanced understanding of the overarching organizational principles of the pulvinar nucleus within the broader context of whole-brain connectivity. By employing cortex-wide correlation analyses of multimodal embedding patterns derived from 'gradients,' which provide spatial maps reflecting the underlying connectomic and molecular similarities across voxels, the study offers a thorough characterization of the functional neuroanatomy of the pulvinar.Weaknesses:Despite its strengths, the current manuscript falls short in presenting the authors' unique perspectives on integrating the diverse biological principles derived from the various neuroimaging modalities. The findings are predominantly reported as correlations between different gradient maps, without providing the in-depth interpretations that would allow for a more comprehensive understanding of the pulvinar's role as a central hub in the brain's network. Another limitation of the study is the lack of clarity regarding the application of pulvinar and its subnuclei segmentation maps to individual brains prior to BOLD signal extraction and gradient reconstruction. This omission raises concerns about the precision and reproducibility of the findings, leaving their robustness less transparently evaluable.

We thank the Reviewer for the valuable comments. While commonalities and discrepancies between structural and functional connectivity have been extensively explored in the literature, the relationship between functional connectivity and modulatory neurotransmission remains poorly understood. Specifically, while the role of thalamic modulatory neurotransmission has been thoroughly investigated in experimental animal models from an electrophysiological perspective, it remains relatively underexplored in the human brain. In our study, we identified significant associations between the spatial distribution of serotonergic, noradrenergic, dopaminergic and mu-opioid systems and functional pulvinar-cortical connectivity to specific functional networks. Evidence from pharmacological challenge studies using resting-state fMRI suggests that these neurotransmission systems may modulate network-specific thalamocortical connectivity directly or influence neural gain in cortico-cortical connectivity, a process partially dependent on thalamocortical connections to associative thalamic nuclei. However, the limitations of spatial and receptor specificity inherent to this approach, coupled with the predominantly correlational nature of our study design, prevented us from drawing more definitive conclusions on the biological relationship between neurotransmitter expression and functional connectivity. As regards the lack of clarity concerning signal extraction, we have now clarified that all the relevant steps of time series extraction were performed in standard space, without any further registration to individual subjects.

**Reviewer #2 (Recommendations for the authors):**
In line with the weaknesses that I raised above, my recommendation to authors are two-fold:(1) Please provide readers with a more holistic viewpoint to better digest all the correlation analyses. For instance, in p18, the summary says:"G_FC_1, GRC1, and G_SC_2 substantially delineate multiscale differences between the ventral and dorsal aspects of the pulvinar. Moving along the ventral-dorsal axis of the pulvinar complex, more ventral regions showed higher functional connectivity to unimodal sensory processing networks, higher levels of 5HTT and NAT expression, and preferentially higher structural connectivity to modality-independent or low-level sensory processing cortices."We already knew somehow the existence of the dorsoventral axis in the pulvinar, as the authors already specified in the introduction. Beyond this simple report on phenomenological observation, one may provide a more integrated discussion to pinpoint what commonality or discrepancy the GFC, GRC, and GSC map show and potential common principles explaining their biological relationship (e.g., the 5HTT and NAT's high expression and functional connectivity). Such digested perspectives will grant the study unique insights into the functional system of the pulvinar.

We have expanded on this topic in the Discussion section (Neurochemical correlates of pulvinar-cortical topographical organization) as follows:

“Indeed, while commonalities and discrepancies between structural and functional connectivity have been extensively investigated, the relationship between functional connectivity and modulatory neurotransmission remains poorly understood. Our findings reveal stronger associations between pulvinar-cortical connectivity to specific functional networks and the spatial distribution of markers of serotonergic, noradrenergic, dopaminergic and opioid systems. Pharmacological challenge studies using resting-state functional MRI suggest that each of these neurotransmission systems may either directly modulate thalamocortical connectivity or influence neuronal gain in cortico-cortical functional connectivity, which is known to depend, in part, on cortical connections to associative thalamic nuclei, including the pulvinar.”

(2) Specify the details if there was a QC procedure to check the signal extraction from the pulvinar subnuclei by applying the segmentation atlas at each individual.

Preprocessed BOLD volumes were available in standard-space, and time series were extracted for each voxel within a standard-space mask of the pulvinar complex. All volumes underwent visual inspection to ensure the accuracy of the registration process. Regarding the pulvinar subnuclei, these structures were not segmented at the individual level.

**Reviewer #3 (Public review):**
Summary of the Study:The authors investigate the organization of the human pulvinar by analyzing DWI, fMRI, and PET data. The authors explore the hypothesis of the "replication principle" in the pulvinar.Strengths and Weaknesses of the Methods and Results:The study effectively integrates diverse imaging modalities to provide a view of the pulvinar's organization. The use of analysis techniques, such as diffusion embedding-driven gradients combined with detailed interpretations of the pulvinar, is a strength.Even though the study uses the best publicly available resolution possible with current MR-technology, the pulvinar is densely packed with many cell bodies, requiring even higher spatial resolution. In addition, the model order selection of gradients may vary with the acquired data quality. Therefore, the pulvinar's intricate organization needs further exploration with even higher spatial resolution to capture gradients closer to the biological organization of the pulvinar.Appraisal of the Study's Aims and Conclusions:The authors delineate the gradient organization of the pulvinar. The study provides a basis for understanding the pulvinar's role in mediating brain network communication.Impact and Utility of the Work:This work contributes to the field by offering insights into pulvinar organization.

We thank the Reviewer for their positive assessment and constructive feedback. The Authors agree with the Reviewer that the spatial resolution of currently available in-vivo imaging methods is limited, and that gradient representation would indeed benefit from higher resolution data. However, we also note that the resolution of structural and functional volumes used in our study is consistent with existing literature on pulvinar connectivity. Additionally, the PET data employed in our work include multi-centric studies collected worldwide from healthy populations, and are primarily acquired using high-resolution scanners that allow spatial resolution up to 2 mm^2^. Notwithstanding, further investigations employing finer resolution imaging techniques, such as ultra-high field fMRI, may provide more detailed insights into pulvinar topographical organization at a finer scale.

**Reviewer #3 (Recommendations for the authors):**
(1) The HCP data contains genetically related datasets. Please mention whether the data-selection criteria for the selected 210 healthy subjects followed the genetically unrelated criteria.

The HCP sample employed in this study consists of an initial cohort of 100 unrelated subjects, as provided in the HCP database, along with an additional random sample of 110 subjects. Subjects were selected without following a genetic criterion, as the family structure of the HCP dataset was part of a restricted access subset that we did not have access to at the time of processing. Subsequently, we obtained access to this information and determined that 178 out of 210 subjects (85%) are genetically unrelated. Of the remaining, genetically related subjects, 22 (~10% of the total sample) were included with another subject from the same family group (11 pairs); 6 (3%) were included with two other family members (2 triplets) and 4 (2%) were all parts of the same family group. This information has been included in the Methods section for clarity.

(2) The study uses HCP data with an fMRI resolution of 2mm isotropic and diffusion MRI with 1.25mm. Additionally, the LEMON dataset includes 1.7mm isotropic DWI data and fMRI with 2.3mm isotropic resolution. Furthermore, the available PET data from the Hansen et al. 2022b study has a rather coarser spatial resolution. Therefore, it may be important to mention in the discussion that the pulvinar is densely packed with cell bodies and that their gradient organization might be better reflected with even higher spatial resolution or improved measurement techniques used in the study.

We have revised the conclusive section of the Discussion into a paragraph title “Future perspectives and limitations”, and added the following text:

“One notable limitation of this study lies in the relatively small size of the pulvinar complex compared to other larger cortical or subcortical structures. The high cellular density of the pulvinar poses a challenge for the relatively coarse resolution of currently available imaging techniques. Although the generally high quality of both the main and validation datasets, including rs-fMRI data (Uǧurbil et al. 2013; Babayan et al. 2019), align with current standards for imaging investigations of pulvinar connectivity, higher-resolution imaging approaches may offer more granular insights. Advanced techniques, such as ultra-high-field fMRI, hold promise for uncovering the fine-scale topographical organization of the pulvinar complex.”

(3) The functional multiplicity of the Pulvinar nuclei among other thalamus nuclei is also illustrated in https://doi.org/10.1038/s42003-022-04126-w

We thank the Reviewer for suggesting this important reference. We have added the following text in the Discussion section:

“It is noteworthy that, in analyses of thalamocortical gradients, the pulvinar complex is situated towards the “sensorimotor” extreme of the unimodal-to-transmodal thalamic gradient (Yang et al., 2020). This likely reflects its prominent connectivity to visual and sensory areas compared to other thalamic nuclei. Nevertheless, the extensive and intricate association of pulvinar with multiple cortical networks emerges is strongly evident in various functional connectivity investigations (Basile et al., 2021; Kumar et al., 2017, 2022). By isolating pulvinar-cortical from broader thalamocortical connectivity, our analysis was able to provide additional insights into the spatial organization of its connectivity with different cortical networks, highlighting the pulvinar's remarkable functional diversity and complexity.”

(4) In addition to DWI/DSI and PET, the study also uses fMRI, which allows for functional interaction in time. It may be worth reflecting in the discussion that the observed gradient organization of the pulvinar could have detailed aspects in the temporal domain, which might not be fully captured in the time-averaged embeddings.

We thank the Reviewer for their insightful observation. The authors recognize that the exploration of brain temporal dynamics is a compelling area of research due to its extensive correlation with multiple hierarchical aspects of brain information processing. Examining the functional organization of the pulvinar complex lies beyond the scope of the present work and will be subject of further investigation. On the other hand, it is possible that certain aspects of the spatial organization of pulvinar connectivity may be influenced by temporal dynamics of cortico-thalamic information processing. Intrinsic timescales have been consistently showed to progressively increase from unimodal to multimodal associative cortical regions. Furthermore, cortico-thalamic connectivity in matrix-rich regions has been correlated with cortical time scales.

To address this point, we have added the following lines to the Discussion section:

“In this context, it could be hypothesized that the observed gradient organization of the pulvinar may also exhibit specific patterns in the temporal domain. Indeed, multiple investigations have linked the temporal dynamics of cortical regions to different aspects of information processing (Rossi-Pool et al., 2021; Soltani et al., 2021). Notably, intrinsic neural timescales of functional activity have been associated with the functional specialization and gradient organization of the cerebral cortex (Golesorkhi et al., 2021), with shorter timescales in unimodal sensory regions and longer ones in transmodal networks (Ito et al., 2020; Murray et al., 2014). Moreover, thalamocortical connectivity has been showed to correlate with these patterns of intrinsic time scale (Müller et al., 2020). In addition, modulatory neurotransmitters such as serotonin and dopamine have been demonstrated to play a significant role in modulating functional cortical dynamics across different timescales (Hansen, Shafiei, et al., 2022; Luppi et al., 2023). Exploring how the spatial organization of the pulvinar relates to temporal dynamics and timescale modulation could provide valuable insights and represents a promising avenue for future investigations.”

(5) The K-means clustering (Supplementary Figure 1) used has limitations, particularly with respect to the structure of the data. Another aspect is the reproducibility of the model-order selection. Did the reliability and reproducibility assessment produce a similar number of clusters with the LEMON data as with the HCP data?

We acknowledge the limitations of k-means clustering, particularly regarding the stability and reproducibility of the model order. To address the concerns, we iteratively ran the clustering algorithm 50 times on bootstrap resamples to enhance the stability of the silhouette score estimates. In addition, we have now replicated the analysis on the secondary dataset, as suggested by the Reviewer (Author response image 2). The Silhouette plots show similar number of clusters between the two different datasets for functional connectivity gradients, with minor differences observed in the results for structural connectivity gradients and multimodal gradient clustering. Notably, we did not find high a high degree of similarity between the results of gradient clustering and histologically defined nuclei, further underscoring the distinct organizational patterns identified through our analysis.

This reinforces the relevance of using gradient-based approaches to reveal insights into the functional and structural organization of the pulvinar complex that may not align strictly with discrete, histologically defined subdivisions.

**Author response image 2. sa3fig2:** K-means clustering of pulvinar gradients on the secondary dataset (LEMON) and their correspondence with histological pulvinar nuclei. Panels on the left show the silhouette plots for left and right pulvinar clustering solutions; error bars are standard error calculated across 50 resamples. Panels on the right show matrix plots of Dice similarity coefficients for pulvinar clusters against histological nuclei (AAL3 atlas). INF: inferior; ANT: anterior; LAT: lateral; MED: medial.

(6) The pulvinar correlates of the unimodal-transmodal cortical gradient (Figure 4) show an association with almost the entire brain (Figure 4C, violin plot). It would be interesting to back this association with known anatomical connectivity studies in animals that show connections to these network areas. To my limited knowledge, I am not aware of pulvinar tracer studies showing such extensive connectivity across the entire cortex.

As our structural connectivity estimates are based on tractography, they are subject to the known limitation of potentially overestimating anatomical connectivity. A technical clarification is warranted: since structural connectivity is grouped by networks, it is strongly influenced by connections to specific cortical regions within each network. This explains the uneven and asymmetric distribution of structural gradient-weighted connectivity observed in our results and does not imply widespread connectivity across the entire cortex.

Nonetheless, structural connectivity of the pulvinar to cortical regions in primates encompasses a remarkably broad array of cortical areas, including predominantly occipital (Adams et al., 2000; Benevento, 1976; Casanova et al., 1989), temporal (Berman & Wurtz, 2010; Gattass et al., 2018; Homman-Ludiye et al., 2020) and parietal cortices (Asanuma et al., 1985; Baleydier & Morel, 1992). Additionally, to a more limited extent, connections to the cingulate gyrus, and portions of the lateral prefrontal cortex have also been documented (Baleydier & Mauguiere, 1985; Baleydier & Mauguire, 1987). These connectivity patterns are in line with prior accounts of structural connectivity of the human pulvinar (Arcaro et al., 2015; Basile et al., 2021; Leh et al., 2008; Tamietto et al., 2012), and with the patterns identified in our work (Author response image 1). Such findings provide further validation of the structural connectivity profiles explored in the present study.

References

Adams, M. M., Hof, P. R., Gattass, R., Webster, M. J., & Ungerleider, L. G. (2000). Visual cortical projections and chemoarchitecture of macaque monkey pulvinar. The Journal of Comparative Neurology, 419(3), 377–393. https://doi.org/10.1002/(SICI)1096-9861(20000410)419:3<377::AID-CNE9>3.0.CO; 2-E

Arcaro, M. J., Pinsk, M. A., & Kastner, S. (2015). The anatomical and functional organization of the human visual pulvinar. Journal of Neuroscience. https://doi.org/10.1523/JNEUROSCI.1575-14.2015

Asanuma, C., Andersen, R. A., & Cowan, W. M. (1985). The thalamic relations of the caudal inferior parietal lobule and the lateral prefrontal cortex in monkeys: Divergent cortical projections from cell clusters in the medial pulvinar nucleus. Journal of Comparative Neurology, 241(3), 357–381. https://doi.org/10.1002/cne.902410309

Baleydier, C., & Mauguiere, F. (1985). Anatomical evidence for medial pulvinar connections with the posterior cingulate cortex, the retrosplenial area, and the posterior parahippocampal gyrus in monkeys. Journal of Comparative Neurology. https://doi.org/10.1002/cne.902320207

Baleydier, C., & Mauguiere, F. (1987). Network organization of the connectivity between parietal area 7, posterior cingulate cortex and medial pulvinar nucleus: A double fluorescent tracer study in monkey. Experimental Brain Research, 66(2). https://doi.org/10.1007/BF00243312

Baleydier, C., & Morel, A. (1992). Segregated thalamocortical pathways to inferior parietal and inferotemporal cortex in macaque monkey. Visual Neuroscience, 8(5), 391–405. https://doi.org/10.1017/S0952523800004922

Basile, G. A., Bertino, S., Bramanti, A., Anastasi, G. P., Milardi, D., & Cacciola, A. (2021). In Vivo Super-Resolution Track-Density Imaging for Thalamic Nuclei Identification. Cerebral Cortex. https://doi.org/10.1093/cercor/bhab184

Benevento. (1976). The Cortical Projections of the Inferior Pulvinar and Adjacent Lateral Pulvinar in the Rhesus Monkey Macaca. October, 108, 1–24.

Berman, R. A., & Wurtz, R. H. (2010). Functional Identification of a Pulvinar Path from Superior Colliculus to Cortical Area MT. The Journal of Neuroscience, 30(18), 6342–6354. https://doi.org/10.1523/JNEUROSCI.6176-09.2010

Cai, L. Y., Yang, Q., Hansen, C. B., Nath, V., Ramadass, K., Johnson, G. W., Conrad, B. N., Boyd, B. D., Begnoche, J. P., Beason-Held, L. L., Shafer, A. T., Resnick, S. M., Taylor, W. D., Price, G. R., Morgan, V. L., Rogers, B. P., Schilling, K. G., & Landman, B. A. (2021). PreQual: An automated pipeline for integrated preprocessing and quality assurance of diffusion weighted MRI images. Magnetic Resonance in Medicine, 86(1), 456. https://doi.org/10.1002/mrm.28678

Casanova, C., Freeman, R. D., & Nordmann, J. P. (1989). Monocular and binocular response properties of cells in the striate-recipient zone of the cat’s lateral posterior-pulvinar complex. Journal of Neurophysiology. https://doi.org/10.1152/jn.1989.62.2.544

Gattass, R., Soares, J. G. M., & Lima, B. (2018). Comparative Pulvinar Organization Across Different Primate Species (pp. 37–37). https://doi.org/10.1007/978-3-319-70046-5_8

Golesorkhi, M., Gomez-Pilar, J., Tumati, S., Fraser, M., & Northoff, G. (2021). Temporal hierarchy of intrinsic neural timescales converges with spatial core-periphery organization. Communications Biology, 4(1), 277. https://doi.org/10.1038/s42003-021-01785-z

Hansen, J. Y., Markello, R. D., Tuominen, L., Nørgaard, M., Kuzmin, E., Palomero-Gallagher, N., Dagher, A., & Misic, B. (2022). Correspondence between gene expression and neurotransmitter receptor and transporter density in the human brain. NeuroImage, 264, 119671. https://doi.org/10.1016/j.neuroimage.2022.119671

Hansen, J. Y., Shafiei, G., Markello, R. D., Smart, K., Cox, S. M. L., Nørgaard, M., Beliveau, V., Wu, Y., Gallezot, J.-D., Aumont, É., Servaes, S., Scala, S. G., DuBois, J. M., Wainstein, G., Bezgin, G., Funck, T., Schmitz, T. W., Spreng, R. N., Galovic, M., … Misic, B. (2022). Mapping neurotransmitter systems to the structural and functional organization of the human neocortex. Nature Neuroscience, 25(11), 1569–1581. https://doi.org/10.1038/s41593-022-01186-3

Homman-Ludiye, J., Mundinano, I. C., Kwan, W. C., & Bourne, J. A. (2020). Extensive Connectivity Between the Medial Pulvinar and the Cortex Revealed in the Marmoset Monkey. Cerebral Cortex, 30(3), 1797–1812. https://doi.org/10.1093/cercor/bhz203

Iglesias, J. E., Insausti, R., Lerma-Usabiaga, G., Bocchetta, M., Van Leemput, K., Greve, D. N., van der Kouwe, A., Fischl, B., Caballero-Gaudes, C., & Paz-Alonso, P. M. (2018). A probabilistic atlas of the human thalamic nuclei combining ex vivo MRI and histology. NeuroImage, 183, 314–326. https://doi.org/10.1016/j.neuroimage.2018.08.012

Ito, T., Hearne, L. J., & Cole, M. W. (2020). A cortical hierarchy of localized and distributed processes revealed via dissociation of task activations, connectivity changes, and intrinsic timescales. NeuroImage, 221, 117141. https://doi.org/10.1016/j.neuroimage.2020.117141

Kumar, V. J., Beckmann, C. F., Scheffler, K., & Grodd, W. (2022). Relay and higher-order thalamic nuclei show an intertwined functional association with cortical-networks. Communications Biology, 5(1), 1–17. https://doi.org/10.1038/s42003-022-04126-w

Kumar, V. J., van Oort, E., Scheffler, K., Beckmann, C. F., & Grodd, W. (2017). Functional anatomy of the human thalamus at rest. NeuroImage, 147, 678–691. https://doi.org/10.1016/j.neuroimage.2016.12.071

Leh, S. E., Chakravarty, M. M., & Ptito, A. (2008). The Connectivity of the Human Pulvinar: A Diffusion Tensor Imaging Tractography Study. International Journal of Biomedical Imaging, 2008, 1–5. https://doi.org/10.1155/2008/789539

Luppi, A. I., Hansen, J. Y., Adapa, R., Carhart-Harris, R. L., Roseman, L., Timmermann, C., Golkowski, D., Ranft, A., Ilg, R., Jordan, D., Bonhomme, V., Vanhaudenhuyse, A., Demertzi, A., Jaquet, O., Bahri, M. A., Alnagger, N. L. N., Cardone, P., Peattie, A. R. D., Manktelow, A. E., … Stamatakis, E. A. (2023). In vivo mapping of pharmacologically induced functional reorganization onto the human brain’s neurotransmitter landscape. Science Advances, 9(24), eadf8332. https://doi.org/10.1126/sciadv.adf8332

Müller, E. J., Munn, B., Hearne, L. J., Smith, J. B., Fulcher, B., Arnatkevičiūtė, A., Lurie, D. J., Cocchi, L., & Shine, J. M. (2020). Core and matrix thalamic sub-populations relate to spatio-temporal cortical connectivity gradients. NeuroImage, 222, 117224. https://doi.org/10.1016/j.neuroimage.2020.117224

Murphy, K., Bodurka, J., & Bandettini, P. A. (2006). How long to scan? The relationship between fMRI temporal signal to noise and necessary scan duration. NeuroImage, 34(2), 565. https://doi.org/10.1016/j.neuroimage.2006.09.032

Murray, J. D., Bernacchia, A., Freedman, D. J., Romo, R., Wallis, J. D., Cai, X., Padoa-Schioppa, C., Pasternak, T., Seo, H., Lee, D., & Wang, X.-J. (2014). A hierarchy of intrinsic timescales across primate cortex. Nature Neuroscience, 17(12), 1661–1663. https://doi.org/10.1038/nn.3862

Oldham, S., & Ball, G. (2023). A phylogenetically-conserved axis of thalamocortical connectivity in the human brain. Nature Communications, 14(1), 6032. https://doi.org/10.1038/s41467-023-41722-8

Rolls, E. T., Huang, C.-C., Lin, C.-P., Feng, J., & Joliot, M. (2020). Automated anatomical labelling atlas 3. NeuroImage, 206, 116189. https://doi.org/10.1016/j.neuroimage.2019.116189

Rossi-Pool, R., Zainos, A., Alvarez, M., Parra, S., Zizumbo, J., & Romo, R. (2021). Invariant timescale hierarchy across the cortical somatosensory network. Proceedings of the National Academy of Sciences, 118(3), e2021843118. https://doi.org/10.1073/pnas.2021843118

Shipp, S. (2003). The functional logic of cortico-pulvinar connections. Philosophical Transactions of the Royal Society B: Biological Sciences, 358(1438), 1605–1624. https://doi.org/10.1098/rstb.2002.1213

Soltani, A., Murray, J. D., Seo, H., & Lee, D. (2021). Timescales of cognition in the brain. Current Opinion in Behavioral Sciences, 41, 30–37. https://doi.org/10.1016/j.cobeha.2021.03.003

Su, J. H., Thomas, F. T., Kasoff, W. S., Tourdias, T., Choi, E. Y., Rutt, B. K., & Saranathan, M. (2019). Thalamus Optimized Multi Atlas Segmentation (THOMAS): Fast, fully automated segmentation of thalamic nuclei from structural MRI. NeuroImage, 194, 272–282. https://doi.org/10.1016/j.neuroimage.2019.03.021

Tamietto, M., Pullens, P., de Gelder, B., Weiskrantz, L., & Goebel, R. (2012). Subcortical Connections to Human Amygdala and Changes following Destruction of the Visual Cortex. Current Biology, 22(15), 1449–1455. https://doi.org/10.1016/j.cub.2012.06.006

Yang, S., Meng, Y., Li, J., Li, B., Fan, Y.-S., Chen, H., & Liao, W. (2020). The thalamic functional gradient and its relationship to structural basis and cognitive relevance. NeuroImage, 218, 116960. https://doi.org/10.1016/j.neuroimage.2020.116960